# Effect of Random Learning Rate: Theoretical Analysis of SGD Dynamics in Non-Convex Optimization via Stationary Distribution

## Abstract

We consider a variant of the stochastic gradient descent (SGD) with a random learning rate and reveal its convergence properties. SGD is a widely used stochastic optimization algorithm in machine learning, especially deep learning. Numerous studies reveal the convergence properties of SGD and its simplified variants. Among these, the analysis of convergence using a stationary distribution of updated parameters provides generalizable results. However, to obtain a stationary distribution, the update direction of the parameters must not degenerate, which limits the applicable variants of SGD. In this study, we consider a novel SGD variant, Poisson SGD, which has degenerated parameter update directions and instead utilizes a random learning rate. Consequently, we demonstrate that a distribution of a parameter updated by Poisson SGD converges to a stationary distribution under weak assumptions on a loss function. Based on this, we further show that Poisson SGD finds global minima in non-convex optimization problems and also evaluate the generalization error using this method. As a proof technique, we approximate the distribution by Poisson SGD with that of the bouncy particle sampler (BPS) and derive its stationary distribution, using the theoretical advance of the piece-wise deterministic Markov process (PDMP).

## 1 Introduction

Stochastic gradient descent (SGD) stands out as a widely employed optimization algorithm in machine learning. It falls under the category of stochastic optimization, where parameters are updated with randomness from the mini-batch sampling. SGD is valued for two main reasons in optimization: (i) it is memory-efficient and requires only low computational resources by updating parameters from a fraction of the training data at each iteration (Bottou, 1991), and (ii) models optimized with SGD have less generalization error than those optimized by other algorithms such as gradient descent (GD) for neural networks (Wu et al., 2020; Zhu et al., 2019). Owing to these advantages, SGD has been one of the standard methods for training deep learning models (Hoffer et al., 2017; Keskar et al., 2016; Zhu et al., 2019).

To understand the properties of SGD, the characteristics of parameters updated by SGD or its variants have been actively studied. As for the usual SGD, Garrigos & Gower (2023) surveyed the results about the convergence rate of SGD in convex and non-convex settings. It also mentions the global convergence property of SGD under the strong convexity setting. Li et al. (2017); Jastrzebski et al. (2017) clarified that the parameter updating process of SGD can be approximated by a stochastic differential equation. Zhu et al. (2019); Nguyen et al. (2019) discussed the relation between the random noise of SGD and the escape efficiency from the sharp minima of the loss function. One example of a variant of SGD is stochastic gradient Langevin dynamics (SGLD), which is an extension of SGD that adds Gaussian noise to the update formula of SGD. Raginsky et al. (2017) analyzed the dynamics of stochastic gradient Langevin dynamics (SGLD) as a variant of SGD and proved the parameters optimized by SGLD converge to the global minima of the generalization error. As another example, Jastrzebski et al. (2017); He et al. (2019); Mandt et al. (2017) analyzed the dynamics of SGD with a constant learning rate under the assumptions that the noise of SGD on the gradient induced by the mini-batch sampling is isotropic, and derived the probability distribution of the parameters obtained by SGD. Latz (2021) analyze SGD both in the case of the constant learning rate and of the decreasing learning rate.

Among the methods analyzing the properties of SGD, one of the most general approaches is to study a *stationary distribution* of parameters updated by SGD and its variants. The stationary distribution is a distribution that remains unchanged when the parameter is updated by one step. It is useful in theoretical analysis, because (i) it can analyze the global dynamics of the optimization algorithm, and (ii) it can be applied to a wide range of loss functions regardless of its shape. For these reasons, we can use it to investigate the optimization of complex loss functions such as those used for training deep neural networks. For example, (Dieuleveut et al., 2020) studied the stationary distribution of the parameter optimized by SGD when the loss function is strongly convex, and (Raginsky et al., 2017) studied the stationary distribution of SGLD when the loss function is non-convex.

Despite the above advantages, there are not many SGD variants to which stationary distribution analysis can be applied. This is because, to use the analysis by a stationary distribution, it is required that the direction of parameter updates by an algorithm does not degenerate; in other words, there must be no directions that are not being explored. Examples of such variants are (i) SGLD (Welling & Teh, 2011; Dalalyan, 2017; Durmus & Moulines, 2016), which adds a Gaussian noise to the parameter update of SGD and (ii) Gaussian SGD (Jastrzebski et al., 2017; He et al., 2019; Mandt et al., 2017), which assumes that the noise of SGD on the gradient induced by the mini-batch sampling is non-degenerate Gaussian. In contrast, the parameter update of SGD degenerates in many practical cases, such as deep learning (Zhu et al., 2019; Nguyen et al., 2019; Simsekli et al., 2019). We remark that we focus on the degeneracy of the update direction of SGD, not on the distribution of it since there is no clear agreement that gradient noise follows a particular distribution (the mathematical definition of degeneracy is in Remark 2). Hence, there is a gap between the variants of SGD considered in the theoretical analysis and the empirical facts about SGD. This gap fosters the following question:

> *Do parameters optimized by a variant of SGD have a stationary distribution*
> *even if the update direction degenerates - and if so, what is the form of it?*

## 1.1 Our Contribution

We theoretically prove that a variant of SGD has a stationary distribution even if the update direction degenerates. Specifically, we develop a novel SGD variant with a *random learning rate*, which follows the Poisson process depending on a mini-batch gradient. We call the variant *Poisson SGD*, and prove that the distribution of a parameter updated by Poisson SGD converges to a stationary distribution. As a result, we provide a positive answer to the question posed above: even with a degenerated parameter update, it is possible to construct a variant of SGD that reaches a stationary distribution by using a random learning rate. We note that our learning rate has a role of efficiently exploring parameters, which differs from conventional methods with adaptive step size such as Adam.

Our specific contributions are as follows. We consider the empirical risk minimization problem and prove the following results under weak assumptions on the loss function such as absolute continuity: (i) the distribution of the parameters updated by Poisson SGD converges to a stationary distribution, and (ii) an output of Poisson SGD converges to the global minima of the empirical risk, applying the stationary distribution while controlling the inverse-temperature parameter. Furthermore, we evaluate the generalization error of the updated parameter for prediction with unseen data by studying an expectation of the risk function in terms of the obtained stationary distribution.

On the technical side, we utilize an algorithm called the Bouncy Particle Sampler (BPS) to demonstrate the convergence to the stationary distribution by Poisson SGD. BPS is a piecewise deterministic Markov process (PDMP) that achieves ergodicity using stochastically occurring jumps (Davis, 1984; 1993). In our proof, we show that the distribution of parameters updated by Poisson SGD can be well approximated by that of BPS, and we concretely construct the stationary distribution using the theory of BPS.

## 1.2 Related Work

There are many works which investigate the stationary distribution of SGD or its variants. Dieuleveut et al. (2020); Chen et al. (2022) derived the stationary distribution of the parameters obtained by SGD when the loss function is strongly-convex, through the theories about Markov processes. The parameters obtained through the SGLD algorithm are theoretically proven to converge to the Gibbs distribution and generalize well (Raginsky et al., 2017). He et al. (2019) and Mandt et al. (2017) assumed the noise of SGD is Gaussian whose covariance matrix is constant and approximate the process of optimization through SGD by Ornstein-Uhlenbeck process and derive its stationary distribution. Gradient Langevin dynamics (GLD), which is a full-batch version of SGLD, can also be seen as a variant of SGD which assumes

that the noise of SGD is Gaussian with a covariance matrix of constant multiples of the identity matrix. Like SGLD, it converges to a stationary distribution even in non-convex scenarios (Dalalyan, 2017; Durmus & Moulines, 2016).

In terms of a random learning rate, there are several empirical studies. Musso (2020) investigated the dynamics of SGD with a random learning rate by analyzing the stochastic differential equation and its Fokker-Planck equation. Blier et al. (2019) showed experimentally that SGD with random learning rates performs well in optimizing deep neural networks. We remark that these studies and ours have several major differences. The first difference is in the design of a learning rate. Our method considers Poisson processes, whereas existing methods consider uniform distributions and heterogeneous learning rates for each subneural network. The second difference is the objective of the study. We aim to evaluate global convergence, while existing studies aim at interpretability, speed of convergence, etc., and have very different motivations.

As for BPS, (Deligiannidis et al., 2019; Durmus et al., 2020) proved that the parameters updated by continuous-time BPS converge to a stationary distribution and derived the concrete form of the stationary distribution and its convergence rate. (Sherlock & Thiery, 2022) clarified the relation between discrete-time BPS and continuous-time BPS.

### 1.3 Notation

For a natural number $a \in \mathbb{N}$, we define $[a] := \{1, 2, ..., a\}$. For a real $z \in \mathbb{R}$, $\lfloor z \rfloor$ denotes the largest integer which is no more than $z$. $I_d$ is a $d$-dimensional identity matrix. $\langle a, b \rangle$ means the inner product in Euclid space, *i.e.*, sum of the product of each component. $\| \cdot \|_1$ and $\| \cdot \|$ mean the vector norms which represent 1-norm and 2-norm respectively. $\mathbb{S}^{d-1}$ is a unit sphere in $\mathbb{R}^d$. For probability measures $P, P'$ on $\mathbb{R}^d$ and $p \in [1, \infty]$, the $p-$Wasserstein distance is defined as $\mathcal{W}_p(P, P') := \inf_{\pi \in \Pi(P,P')} (\int_{\mathbb{R}^d} \|z - z'\|_p^p d\pi(z, z'))^{1/p}$, where $\Pi(P, P')$ is a set of coupling measure between $P$ and $P'$. $\|P - P'\|_{\mathrm{TV}}$ denotes the total variation of $P - P'$. $\Gamma : \mathbb{R} \to \mathbb{R}$ denotes the gamma function, *i.e.*, $\Gamma(z) = \int_0^\infty t^{z-1} e^{-t} dt$. $\mathrm{B} : \mathbb{R} \times \mathbb{R} \to \mathbb{R}$ denotes the beta function, *i.e.*, $\mathrm{B}(x, y) = \int_0^1 t^{x-1}(1-t)^{y-1} dt$. For a compact set $\Theta$, we denote $\mathrm{diam}(\Theta) = \sup_{\theta_1, \theta_2 \in \Theta} \|\theta_1 - \theta_2\|$. For a random variable $X \in \mathcal{X}$, $\mathbb{E}_X[X]$ denotes the expected value with regard to $X$, *i.e.*, $\int_{\mathcal{X}} x d\mu_X(dx)$, where $\mu_X$ is the probability measure of $X$. $\mathbb{1}[\cdot]$ denotes an indicator function, which takes 1 if the condition in the bracket is satisfied, and 0 otherwise. We denote $a_+ = \max\{0, a\}$.

## 2 Preliminary

### 2.1 Problem Setup: Empirical Risk Minimization

We consider the following stochastic optimization problem. Let $\mathcal{Z}$ be a compact sample space, and consider a probability measure $P_*$ on $\mathcal{Z}$. Suppose that we observe $n$ samples $\mathbf{z} = \{z_1, ..., z_n\} \subset \mathcal{Z}$, that are independently and identically generated from the measure $P_*$. Using the samples, we consider an empirical risk with a loss function. Let $\Theta \subset \mathbb{R}^d$ be a compact parameter space, and define $W := \mathrm{diam}(\Theta)$. With a (potentially non-convex) loss function $\ell : \mathcal{Z} \times \Theta \to \mathbb{R}$, we consider the following empirical risk with the samples:

$$L_{\mathbf{z}}(\theta) = \frac{1}{n} \sum_{i=1}^n \ell(z_i; \theta), \ \theta \in \Theta. \tag{1}$$

Our goal is to find a global minimum of the empirical risk $L_{\mathbf{z}}(\cdot)$, which is defined as a parameter $\theta^* \in \Theta$ which satisfies

$$L_{\mathbf{z}}(\theta^*) = \min_{\theta' \in \Theta} L_{\mathbf{z}}(\theta'). \tag{2}$$

In this study, we consider a torus as the parameter space $\Theta$ in order to skip the argument of constraining parameter updates to a compact set. If we consider another compact parameter space such as a hypercube, a projection onto the space needs to be added. We also mention that the boundedness of parameter spaces is an accepted setting in previous studies on SGD, e.g. Ljung (1977); Kushner & Yin (2003); Bonnabel (2013); Tripuraneni et al. (2018); Lan (2020); Boumal (2023).

### 2.2 Gradient Descent Algorithm and Variants

To find the global minimum $\theta^*$ as defined in (2), we often use the optimization algorithm called stochastic gradient descent (SGD) with momentum.

#### 2.2.1 General Form of Stochastic Gradient Descent

We give a formal definition of SGD with a momentum term associated with empirical risk $L_{\mathbf{z}}(\theta)$ in (1). Let $K \in \mathbb{N}$ be the number of iterations. The SGD with momentum generates a sequence of $\Theta$-valued random parameters $\theta_1, ..., \theta_K$ and $\mathbb{R}^d$-valued random vectors $v_1, ..., v_K$, by the following procedure.

Let $\theta_0 \in \Theta$ be an arbitrary parameter for the initialization, $v_0 \in \mathbb{R}^d$ as an initial velocity vector, and $m \in [n]$ be a number of sub-samples, i.e., the batch size. Suppose that we observe the $n$ samples $z := \{z_1, ..., z_n\}$, i.e., the full-batch. For $k = 1, ..., K$, we uniformly sample $m$ integers $I^{(k)} = \{i_1, i_2, ..., i_m\}$ from $[n]$, which is called mini-batch sampling with the batch-size $m$. We define an associated mini-batch risk as

$$\widehat{L}_{\mathbf{z}}^{(k)}(\theta) := \frac{1}{m} \sum_{i \in I^{(k)}} \ell(z_i; \theta). \tag{3}$$

Then, with initial values $\theta_0 \in \Theta$ and $v_0 \in \mathbb{R}^d$, the SGD with momentum generates the parameter and the velocity vector by the following recursive formula for $k = 1, ..., K$:

$$\theta_k = \theta_{k-1} + \eta_k v_{k-1}, \text{ and}$$
$$v_k = v_{k-1} - \alpha_k \nabla \widehat{L}_{\mathbf{z}}^{(k)}(\theta_k), \tag{4}$$

where $\eta_k > 0$ is a learning rate and $\alpha_k \in \mathbb{R}$ is a momentum coefficient. This form is generic and can be identical to other forms of SGD with momentum (Qian, 1999; Sutskever et al., 2013) by adjusting the parameters $\eta$ and $\alpha$.

**Remark 1** (Gradient Noise). For the sake of technical discussions below, we define a notion of *gradient noise* $\xi_k^{(m,n)}(\theta) := \nabla \widehat{L}_{\mathbf{z}}^{(k)}(\theta) - \nabla L_{\mathbf{z}}(\theta)$ for $k = 1, ..., N$ and $\theta \in \Theta$, which is caused by sub-sampling of the SGD. If one assumes that $\xi_k^{(m,n)}(\theta)$ follows a centered Gaussian distribution with an identity covariance, the SGD corresponds to the gradient Langevin dynamics (GLD). However, it is empirically observed that the covariance matrix of the gradient noise often degenerates (Zhu et al., 2019; Cheng et al., 2020; HaoChen et al., 2021). In addition, there is still much discussion on a distribution that gradient noise follows, e.g. Simsekli et al. (2019) and Battash et al. (2024) reports the non-Gaussianity of the gradient noise in empirical studies. Due to these situations, we do not consider a full-rank covariance matrix nor a particular distribution of the gradient noise.

**Remark 2** (Degeneracy of the gradient noise). We briefly explain the *degeneracy* of the gradient noise. Since the covariance matrix of the gradient noise with the batch-size $m$ is written as

$$\frac{1}{m} \sum_{i=1}^{m} (\nabla \ell(z_i; \theta) - \nabla L_{\mathbf{z}}(\theta))(\nabla \ell(z_i; \theta) - \nabla L_{\mathbf{z}}(\theta))^\top,$$

where each term in the sum is a rank-1 matrix, the rank of a total covariance matrix is no greater than $m$. Hence, in the over-parameterized models like neural networks, the matrix becomes rank-deficient, which we refer to as degeneracy of the noise.

## 3 Our SGD Variant: Poisson SGD

In this section, we introduce our algorithm, *Poisson SGD*, which is a variant of SGD with a random learning rate $\eta$ and momentum coefficient $\alpha$. We design our method so that the parameter can search the whole parameter space owing to the design.

We describe the random learning rate. In preparation, we define the following exponential distribution function with a function $f : \Theta \to \mathbb{R}^d$ and parameters $\theta \in \Theta, v \in \mathbb{S}^{d-1}$:

$$E(f(\cdot), \theta, v) := \exp\left(-\int_0^t \{\max\{\langle f(\theta + rv), v \rangle, 0\} + C_P\} dr\right),$$

---

**Algorithm 1** Poisson SGD

1: Initialize $(\theta_0, v_0)$ as $\|v_0\| = 1$.
2: **for** $k = 1, 2, ..., K$ **do**
3:     Sample $I^{(k)} \subset [n]$ and obtain $\nabla \widehat{L}_{\mathbf{z}}^{(k)}(\theta_k)$ as (3).
4:     Sample $\eta_k$ as (5).
5:     Obtain $\theta_k$ as $\theta_k = \theta_{k-1} + \eta_k v_{k-1}$.
6:     Obtain $v_k$ as $v_k = v_{k-1} - \alpha_k \nabla \widehat{L}_{\mathbf{z}}^{(k)}(\theta_k)$ with $\alpha_k$ as (6).
7: **end for**
8: Return $(\theta_K, v_K)$.

---

where $C_P > 0$ is some constant. Then, for each update $k = 1, ..., K$, we design the random learning rate $\eta_k$ following the exponential distribution:

$$P(\eta_k \geq t) = E(\beta \nabla \widehat{L}_{\mathbf{z}}^{(k)}(\cdot), \theta_{k-1}, v_{k-1}), \tag{5}$$

where $\beta > 0$ is the hyper-parameter of Poisson SGD, called an inverse temperature parameter.

Second, we select the momentum coefficient $\alpha_k$ for each $k = 1, ..., K$ as

$$\alpha_k = 2 \frac{\langle \nabla \widehat{L}_{\mathbf{z}}^{(k)}(\theta_k), v_{k-1} \rangle}{\|\nabla \widehat{L}_{\mathbf{z}}^{(k)}(\theta_k)\|^2} + C_\alpha, \tag{6}$$

where $C_\alpha \geq 0$ is the hyper-parameter. While $C_\alpha$ has the function of enhancing the effect of the gradient for practical use, we set $C_\alpha = 0$ in the theoretical analysis of this paper (in experiments in Section 7, we set $C_\alpha > 0$). This setup keeps the length of the velocity vector constant as $\|v_k\| = 1$ for every $k$ (See Proposition 7 in Appendix), and only uses its angle to update the parameters. We update the parameter by changing $\eta_k$ and $\alpha_k$ in every iteration. The pseudo-code of Poisson SGD is shown in Algorithm 1.

The algorithm is designed to effectively explore large regions of the parameter space $\Theta$. Specifically, the update direction is determined by the velocity vector $v_k$ normalised by $\alpha_k$ as (6), and the size of the update is randomly set by the random learning rate $\eta_k$ as (5). When the gradient $\nabla \widehat{L}_{\mathbf{z}}^{(k)}(\cdot)$ is small, the learning rate $\eta_k$ is chosen to be large, thus the updated parameter tends to escape local minima or saddle points. Figure 1 illustrates that Poisson GD, which we refer to as the full-batch version of Poisson SGD explores a wider parameter space and discovers the global minimum owing to the random learning rate, while the parameters updated by GD converge to the local minimum. Here, we set the learning rate of GD as $\eta = 0.02$ and the hyper-parameter of Poisson GD as $C_P = 100$ and $\beta = 10000$.

**Remark 3** (Moments of Poisson SGD). We claim that even if the learning rate is random, the actual updates are not too large, by studying its moments. That is, if $C_P$ is sufficiently large, there is little chance of sampling a large learning rate $\eta$, since the first and second moments of $\eta_k$ are given as $\mathbb{E}[\eta_k] \leq \int_0^\infty \exp(-C_P t) dt = \frac{1}{C_P}$ and $\mathbb{E}[\eta_k^2] - \mathbb{E}[\eta_k]^2 \leq \int_0^\infty 2s \exp(-C_P s) ds - \frac{1}{C_P^2} = \frac{1}{C_P^2}$. By this property, we can avoid the case in which $\eta_k$ diverges. In addition, even if a large $\eta_k$ is sampled, the parameter does not exit from the parameter space since we consider a torus as the parameter space.

## 4 Convergence Theory for Poisson SGD

We provide theoretical results on the convergence of Poisson SGD (Algorithm 1). Our main interest is a distribution of the generated parameter $\theta_K$ by Poisson SGD associated with the empirical risk minimization problem (2).

### 4.1 Stationary Distribution of Poisson SGD

In this section, we show that the parameter $\theta_K$ by the Poisson SGD follows a stationary distribution. Formally, we define the stationary distribution of the Markov process. In preparation, we utilize the notion of *transition probability* $Q(\theta, dw)$ from a distribution $p_0(\theta)$ to another $p_1(\theta)$ on $\Theta$, that is, $p_1(w) = \int_\Theta Q(\theta, dw) p_0(d\theta)$ holds.

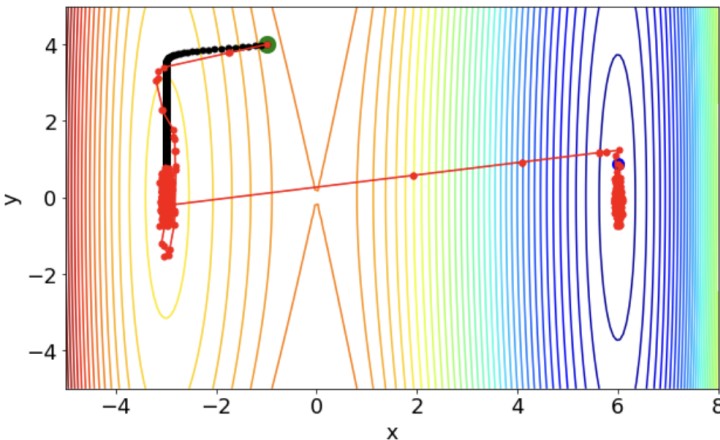

Figure 1: The comparison of the trajectories of GD (with a fixed learning rate) and Poisson GD (with the random learning rate) in optimizing the function $z = x^4 - 4x^3 - 36x^2 + y^2$. Poisson GD represents replacing the mini-batch loss $\widehat{L_{\mathbf{z}}}$ by the full-batch loss $L_{\mathbf{z}}$ in Poisson SGD, where we set $L_{\mathbf{z}}(x, y) = x^4 - 4x^3 - 36x^2 + y^2$. The point $(-3, 0)$ represents a local minimum and the point $(6, 0)$ is identified as the global minimum. A green point indicates the initial position, a black line represents the trajectory of GD, and a red line represents the trajectory of Poisson GD.

**Definition 1.** Let $Q(\theta, dw)$ be the transition probability of a Markov process in $\Theta$. If the following equation holds, we call the probability distribution $\pi(\theta)$ stationary distribution of the Markov process:

$$\pi(dw) = \int_{\Theta} Q(\theta, dw)\pi(d\theta).$$

A stationary distribution is an useful notion to represent a limit of the parameter distribution, and it enables us to analyze where the parameter converges by algorithms. For example, see the theoretical framework to analyze stochastic optimization algorithms by (Raginsky et al., 2017).

### 4.1.1 Assumption

We provide several principal assumptions. First, we consider the basis assumptions on the loss function $\ell(\cdot; \cdot)$. The following conditions are fairly general for the analysis of stochastic optimization algorithms, e.g. Bertazzi et al. (2022).

**Assumption 1** (Loss function). *The loss function $\ell : \mathcal{Z} \times \Theta \to \mathbb{R}_{\geq 0}$ satisfies the following conditions:*

- *$\ell(z; \theta)$ is absolutely continuous and differentiable with respect to $\theta \in \Theta$ for every $z \in \mathcal{Z}$.*
- *$\nabla_{\theta}\ell(z; \theta)$ is continuous in $\theta$ and $z$ for all $\theta \in \Theta$ and $z \in \mathcal{Z}$.*

The first condition is satisfied by a large class of models, such as linear regression model, or deep neural networks whose activation function is ReLU or sigmoid function. From the second condition, we define an upper bound $M_{\ell} := \max_{\theta \in \Theta, z \in \mathcal{Z}} \|\nabla_{\theta}\ell(z; \theta)\|$ since $\Theta$ and $\mathcal{Z}$ are compact.

### 4.1.2 Statement of Convergence

Let $\mu_{\mathbf{z}, K}$ be a distribution of the output $\theta_K$ from the Poisson SGD in Algorithm 1 with the given dataset $\mathbf{z}$. We discuss the convergence of $\mu_{\mathbf{z}, K}$ as $K$ increases.

In preparation, we define a probability measure on $\Theta$ for arbitrary $\beta, \varepsilon > 0$, whose density is written as follows:

$$\mu_{\mathbf{z}}^{(\beta, \varepsilon)}(d\theta) \propto \left(\beta M_{\ell} + \frac{1}{\varepsilon} + a_d \beta \|\nabla L_{\mathbf{z}}(\theta)\|\right) \exp(-\beta L_{\mathbf{z}}(\theta))d\theta, \tag{7}$$

where $a_d := \Gamma(d/2)/(\sqrt{\pi}\Gamma(d/2 + 1/2))$. The probability measure (7) is concentrated around the global minima of $L_{\mathbf{z}}(\theta)$, since the dominant exponential term $\exp(-\beta L_{\mathbf{z}}(\theta))$ in (7) increases in $L_{\mathbf{z}}(\theta)$. In addition, as the inverse temperature parameter $\beta$ increases, the measure $\mu_{\mathbf{z}}^{(\beta,\varepsilon)}$ concentrates more around the global minimum.

We show our results on the convergence of the stationary distribution. The discrepancy is measured by the Wasserstein distance $\mathcal{W}_1(\cdot, \cdot)$. We remark that this theorem is the integration of Theorem 3 and Theorem 4 appearing later in Section 5. Recall that we defined $W := \text{diam}(\Theta)$.

**Theorem 1** (Stationary distribution of Poisson SGD). *Fix arbitrary $\beta, \varepsilon > 0$. Suppose Assumption 1 holds. We set the hyper-parameter of Poisson SGD as $C_P = 1/\varepsilon$. Then, for any $K \in \mathbb{N}$, the distribution $\mu_{z,K}$ satisfies*

$$\mathcal{W}_1(\mu_{z,K}, \mu_z^{(\beta,\varepsilon)}) \le 4\sqrt{d}K\varepsilon + W \cdot \kappa(\beta, \varepsilon, d)^K, \tag{8}$$

*where $0 < \kappa(\beta, \varepsilon, d) < 1$ is a constant.*

*Moreover, if $\kappa(\beta, \varepsilon, d)$ satisfies $\lim_{K \to \infty} \kappa(\beta, \delta/K, d)^K = 0$ with some $\delta > 0$, there exists a sequence $\varepsilon = \varepsilon_K \searrow 0$ as $K \to \infty$ such that $\mathcal{W}_1(\mu_{z,K}, \mu_z^{(\beta,\varepsilon)}) = o(1)$ as $K \to \infty$ holds.*

This theorem shows that the parameter distribution $\mu_{\mathbf{z},K}$ by Poisson SGD converges to the stationary distribution $\mu_{\mathbf{z}}^{(\beta,\varepsilon)}$ owing to the random learning rate (5). This is contrast to ordinary SGD, which is not shown to converge to a stationary distribution. Further, Poisson SGD does not make any assumptions on the gradient noise $\xi_k^{(n,m)}$ in Remark 1, unlike SGLD, which converges to a stationary distribution by introducing Gaussianity in the gradient noise.

The right-hand side in (8) shows an approximation-complexity trade-off of Poisson SGD described as follows. In preparation, we will introduce a certain stochastic process to achieve the stationary distribution $\mu_{\mathbf{z}}^{(\beta,\varepsilon)}$ (detail is in Section 5). The first term of (8) describes an approximation error of Poisson SGD to the stochastic process. The second term of (8) denotes a convergence error of the stochastic process to the stationary distribution $\mu_{\mathbf{z}}^{(\beta,\varepsilon)}$, which reflects the complexity of the stochastic process. $\varepsilon$ is a parameter for the stochastic process and controls the balance between the approximation error and the complexity error.

We further discuss the additional assumption $\lim_{K \to \infty} \kappa(\beta, \delta/K, d)^K = 0$. This condition is related to the convergence rate of the approximated stochastic process of Poisson SGD. Although the explicit form of $\kappa(\beta, \delta/K, d)$ is not clarified in our case, there is a common example having its explicit form. One example is SGLD: Raginsky et al. (2017) shows that a form of $\kappa(\beta, \delta/K, d)$ can be calculated, because SGLD is reduced to the Langevin process.

**Remark 4** (Form of $\kappa(\beta, \varepsilon, d)$). We discuss a form of $\kappa(\beta, \varepsilon, d)$ of other related algorithms, although we could not achieve the explicit form of $\kappa(\beta, \varepsilon, d)$ of Poisson SGD. In the case of Langevin dynamics with the setting of Raginsky et al. (2017), $\kappa(\beta, \varepsilon, d)$ is $\Omega(c_{LS}k\eta/\beta(\beta + d))$, where $c_{LS}$ is the logarithmic Sobolev constant. On the other hand, explicitly deriving $\kappa(\beta, \varepsilon, d)$ for a class of PDMP is a challenging task as described in Deligiannidis et al. (2019); Durmus et al. (2020), as well as that of Poisson-SGD.

**Remark 5** (Comparison with SGLD). We discuss the difference between Poisson SGD and SGLD, which is another method achieving a stationary distribution. First, while SGLD adds a Gaussian noise to the update formula of SGD, Poisson SGD does not have an additive noise. The second difference is the form of the stationary distribution. A stationary distribution of SGLD is the Gibbs distribution, and that of Poisson SGD has the different form (7). This difference is derived from the random learning rate of Poisson SGD.

**Remark 6** (Relation to flat minima). From Theorem 1, we can state the property of Poisson SGD being easier to go to the flat minima than the sharp minima. We consider the probability of existence around a flat minimum $\theta_1 \in \Theta$ and a sharp minimum $\theta_2 \in \theta$, when we find that, due to the shape of the distribution, a measure of an $\epsilon$-neighborhood of $\theta_1$ is greater than that within an $\epsilon$-neighborhood of $\theta_2$. Hence, we can claim that Poisson SGD also tends to favor flat minima.

### 4.2 Global Convergence

We discuss the global convergence statement, that is, the empirical risk $L_{\mathbf{z}}(\theta_K)$ with Poisson SGD is minimized with high probability. We consider the additional assumption for the loss function $\ell$:

**Assumption 2.** *With some $c_1 > 0$, $\sup_{z \in \mathcal{Z}} \|\nabla\ell(z; \theta_1) - \nabla\ell(z; \theta_2)\| \le c_1\|\theta_1 - \theta_2\|$ holds for every $\theta_1 \ne \theta_2 \in \Theta$.*

Then, we obtain the following global convergence theorem. We define $B := \sup_{z \in \mathcal{Z}} \|\nabla \ell(z; 0)\|$ by following Assumption 1.

**Theorem 2** (Global convergence of Poisson SGD on empirical risk). *Fix arbitrary $\beta, \varepsilon > 0$. Consider Poisson SGD in which $C_P = 1/\varepsilon$. Let the upper bound of $\mathcal{W}_1(\mu_{z,K}, \mu_z^{(\beta,\varepsilon)})$ obtained in Theorem 1 be $d_K(\beta, \varepsilon, d)$. Also, suppose that Assumption 1 and 2 hold. Then, it holds that*

$$
\mathbb{E}_{\theta_K \sim \mu_{z,K}}[L_z(\theta_K)] - \min_{\theta \in \Theta} L_z(\theta)
$$

$$
\leq (c_1 W + B)\sqrt{W d_K(\beta, \varepsilon, d)} + \frac{1}{\beta}\left(\frac{d}{2}\log\frac{eW^2 c_1 \beta}{d} + \log\left(1 + \frac{a_d(c_1 W + B)}{M_\ell}\right)\right). \tag{9}
$$

Theorem 2 states that we can make $\mathbb{E}[L_{\mathbf{z}}(\theta_K)]$ be arbitrarily close to $\min_{\theta \in \Theta} L_{\mathbf{z}}(\theta)$ by selecting large $\beta$, provided that we can make $d_K(\beta, \varepsilon, d)$ arbitrarily small by the choice of $\varepsilon$ and $K$ in spite of $\beta$. Intuitively, Poisson SGD achieves global convergence by appropriately adjusting the learning rate and momentum coefficient based on the shape of the loss function at the current location. Poisson SGD achieves the global convergence by the similar approach of global convergence of SGLD by Raginsky et al. (2017).

The right-hand side of (9) is divided into two terms. The first term expresses the distance between the parameter and its stationary distribution. The second represents the degree of concentration of the stationary distribution $\mu_{\mathbf{z}}^{(\beta,\varepsilon)}$ on the global optima. The higher the inverse temperature $\beta$, the more the term decreases.

## 5 Proof Outline

### 5.1 Overview

We give an overview of a proof of Theorem 1. In preparation, we present several key concepts: (i) the property of the *piece-wise deterministic Markov process* (PDMP) (Davis, 1984; 1993), and (ii) the ergodicity of *bouncy particle sampler* (BPS) (Peters & de With, 2012). The PDMP is a class of Markov processes that behave deterministically for some period and jumps randomly, which easily converges to a stationary distribution. BPS is a stochastic algorithm in the class of the PDMP.

We show the statement by the following steps:

(I) We show that the distribution of the parameter by Poisson SGD is sufficiently close to that of a parameter by BPS. We show this claim by using the approximation theory on PDMP (Theorem 3).

(II) We derive a stationary distribution and the ergodicity of BPS, following previous researches (Theorem 4).

### 5.2 Design of BPS

We introduce BPS, which is one of the most popular algorithms in PDMPs, and actively studied in terms of MCMC algorithm (Deligiannidis et al., 2019; Bouchard-Côté et al., 2018). BPS generates a sequence of parameters $\{\widehat{\theta}_k\}_{k=1}^K \subset \Theta$ and velocity vectors $\{\widehat{v}_k\}_{k=1}^K \subset \mathbb{R}^d$ in its recursive manner, as shown in Algorithm 2. Let $(\widehat{\theta}_0, \widehat{v}_0)$ be the initialization. For the $k$-th iteration, BPS generates a learning rate $\widehat{\eta}_k$ from an exponential distribution whose intensity depends on the previous pair $(\widehat{\theta}_{k-1}, \widehat{v}_{k-1})$ and the positive constants $\Lambda_{\mathrm{ref}}$ and $C_B$. After obtaining the parameter $\widehat{\theta}_k$, we consider the stochastic update of the velocity vector. That is, with the probability

$$
\widehat{p}_k := \frac{\beta\langle \nabla L_{\mathbf{z}}(\widehat{\theta}_k), \widehat{v}_{k-1}\rangle_+ + C_B}{\beta\langle \nabla L_{\mathbf{z}}(\widehat{\theta}_k), \widehat{v}_{k-1}\rangle_+ + \Lambda_{\mathrm{ref}} + C_B}, \tag{10}
$$

we update the velocity vector with the gradient of the full-batch loss $\nabla L_{\mathbf{z}}$, otherwise with the sample from the uniform distribution on $\mathbb{S}^{d-1}$. The former update is called *reflection*, and the latter is *refreshment*. We remark that $\|\widehat{v}_k\|$ is constant for $k = 1, 2, ..., K$ in the same way as Poisson SGD (See Proposition 7 in Appendix).

### 5.3 Connect Poisson SGD and BPS

We show that the output distribution of Poisson SGD and that of BPS are sufficiently close as in the following statement:

---

**Algorithm 2** Bouncy Particle Sampler

---

1: Initialize $(\widehat{\theta}_0, \widehat{v}_0)$ as $\|\widehat{v}_0\| = 1$.
2: **for** $k = 1, 2, ..., K$ **do**
3:     Sample $\widehat{\eta}_k$ as $\widehat{\eta}_k \sim P(\widehat{\eta}_k \geq t) = \exp\left(-\int_0^t \{\beta\langle\nabla L_{\mathbf{z}}(\widehat{\theta}_{k-1} + r\widehat{v}_{k-1}), \widehat{v}_{k-1}\rangle_+ + \Lambda_{\text{ref}} + C_B\}dr\right)$
4:     Update $\widehat{\theta}_k$ as $\widehat{\theta}_k = \widehat{\theta}_{k-1} + \widehat{\eta}_k\widehat{v}_{k-1}$
5:     With probability $\widehat{p}_k$ as (10), update $\widehat{v}_k$ as

$$\widehat{v}_k = \widehat{v}_{k-1} - 2\frac{\langle\nabla L_{\mathbf{z}}(\widehat{\theta}_k), \widehat{v}_{k-1}\rangle}{\|\nabla L_{\mathbf{z}}(\widehat{\theta}_k)\|^2}\nabla L_{\mathbf{z}}(\widehat{\theta}_k)$$

    Otherwise, update $\widehat{v}_k$ as
$$\widehat{v}_k \sim \text{Unif}(\mathbb{S}^{d-1})$$

6: **end for**
7: Return $(\widehat{\theta}_K, \widehat{v}_K)$

---

**Theorem 3** (Distance between Poisson SGD and BPS). *Fix arbitrary $\beta, \varepsilon > 0$. As for Poisson SGD, we set $C_P = 1/\varepsilon$. As for BPS, we set $\Lambda_{\text{ref}}$ and $C_B$ as $\Lambda_{\text{ref}} + C_B = \beta M_\ell + 1/\varepsilon$. Let the distribution of the obtained parameter by Poisson SGD and BPS be $\mu_{z,K}$ and $\widehat{\mu}_{z,K}$ respectively. We set the same initial value between Poisson SGD and BPS. Then, the following holds:*

$$\mathcal{W}_1(\mu_{z,K}, \widehat{\mu}_{z,K}) \leq 4\sqrt{d}K\varepsilon.$$

For proving this theorem, we calculate the distance between Poisson SGD and BPS by a one-step update. Then, we simply accumulate this error for $K$ times. In this discussion, we mainly use the property that if learning rate $\eta_k$ and $\widehat{\eta}_k$ are small, the difference of $v_k$ and $\widehat{v}_k$ is also made to be small. This type of discussion is also used in Raginsky et al. (2017).

### 5.4 The Stationary Distribution and Ergodicity of BPS

In this section, we investigate the stationary distribution and ergodicity of BPS. First, we define the term *ergodicity*.

**Definition 2** (Ergodicity). We consider the discrete-time Markov process. If the process converges to a unique stationary distribution, we call the process has the ergodicity. Especially, if the ergodic process converges to its stationary distribution by the exponential rate about the number of iteration $k$, the process is called exponentially ergodic.

Without ergodicity, the stochastic process may converge to more than one stationary distribution, or not converge to any stationary distribution due to stacking to a saddle point in the parameter space. So we have to prove this property when we try to analyze the stationary distribution of a stochastic process.

Now, we show our result about BPS.

**Theorem 4** (Stationary Distribution of BPS). *Suppose that Assumption 1 holds. Set the parameter of BPS, $\Lambda_{\text{ref}}$ and $C_B$ as in Theorem 3. Then, the distribution $\widehat{\mu}_{z,K}$ of the obtained parameters $\widehat{\theta}_K$ by BPS satisfies the following:*

$$\|\widehat{\mu}_{z,K} - \mu_z^{(\beta,\varepsilon)}\|_{\text{TV}} \leq \kappa(\beta, \varepsilon, d)^K,$$

*where $\kappa(\beta, \varepsilon, d)$ is a positive constant less than 1.*

In the proof of this theorem, we use the discussion in Deligiannidis et al. (2019) which showed that continuous-time BPS converges to the unique stationary distribution $\pi(\theta) \propto \exp(-U(\theta))$ by the exponential rate in TV distance.

## 6 Generalization Error Analysis

We define an expected risk of $\theta \in \Theta$, also known as the generalization error

$$L(\theta) := \mathbb{E}_{z\sim P_*}[\ell(z;\theta)],$$

which measures a prediction performance with unseen data. We calculate the generalization error of the parameter obtained by the Poisson SGD, using the discussion in Raginsky et al. (2017).

Now, we give our results. We define $A := \sup_{z \in \mathcal{Z}} |\ell(z; 0)|$ by following Assumption 1.

**Theorem 5** (Generalization Error of Poisson SGD). *Suppose that Assumption 1 and 2 hold. Let $\theta_K$ be the parameter obtained by Poisson SGD with $C_P = 1/\varepsilon$. Then, we obtain the following bound:*

$$\mathbb{E}_{z \sim P_*^n}[\mathbb{E}_{\theta_K \sim \mu_{z,K}}[L(\theta_K)]] - \min_{\theta \in \Theta} L(\theta)$$

$$\leq (c_1 W + B) \left( \sqrt{W d_K(\beta, \varepsilon, d)} + 2W \left( \left( \frac{C_d + \beta C}{n} \right)^{\frac{1}{2}} + \left( \frac{C_d + \beta C}{n} \right)^{\frac{1}{4}} \right) \right)$$

$$+ \frac{1}{\beta} \left( \frac{d}{2} \log \frac{e W^2 c_1 \beta}{d} + \log \left( 1 + \frac{a_d(c_1 W + B)}{M_\ell} \right) \right),$$

*where $d_K(\beta, \varepsilon, d)$ is the upper bound of the Wasserstein distance in Theorem 1, $C_d = 4a_d(c_1 W + B)/M_\ell$, and $C = c_1 W^2 + 2BW + 2A$.*

Theorem 5 states that the expected value of the generalization error of Poisson SGD can be arbitrarily close to its global optima in $\theta \in \Theta$, by selecting small $\varepsilon$, large $K$, large $\beta$, and large $n$, provided that $d_K(\beta, \varepsilon, d)$ can be arbitrarily small only by the choice of $\varepsilon$ and $K$.

We further discuss a way of improve an order of the generalization bound in Theorem 5. While our bound has the order $O((1/n)^{1/4})$, we can obtain an order $O(1/n)$ by using the *dissipativity* condition of the loss function, which is used in Raginsky et al. (2017) for SGLD. The dissipativity condition allows us to derive log-Sobolev inequality for $L_{\mathbf{z}}(\theta)$, which leads the improved sample complexity. We state this fact in the following proposition.

**Proposition 6.** *Suppose that the same condition and setting as Theorem 5 hold. In addition, we assume that the Gibbs distribution $\nu_{\mathbf{z}}^{(\beta)} \propto \exp(-\beta L_z(\theta))$ satisfies the log-Sobolev inequality for any dataset $\mathbf{z} = \{z_1, ..., z_n\}$, that is,*

$$\mathbb{E}[f(\theta)^2 \log f(\theta)^2] - \mathbb{E}[f(\theta)^2] \log \mathbb{E}[f(\theta)^2] \leq c_{LS}^{(\beta)} \mathbb{E}[\|\nabla f(\theta)\|^2]$$

*holds for all smooth functions $f$ and any data $\mathbf{z} = \{z_1, ..., z_n\}$, where $\theta \sim \nu_{\mathbf{z}}^{(\beta)}$ and $c_{LS}^{(\beta)} < \infty$ is a constant. Then, the following holds:*

$$\mathbb{E}_{z \sim P_*^n}[\mathbb{E}_{\theta_K \sim \mu_{z,K}}[L(\theta_K)]] - \min_{\theta \in \Theta} L(\theta)$$

$$\leq (c_1 W + B) \left( \sqrt{W d_K(\beta, \varepsilon, d)} + \frac{2c_{LS}^{(\beta)} \beta M_\ell}{n} \right) + W \sqrt{2c_{LS}^{(\beta)} \log(1 + a_d \beta \varepsilon M_\ell)} + \frac{d}{2\beta} \log \left( \frac{e W^2 c_1 \beta}{d} \right).$$

# 7 Experiments

We give several experimental result to validate our theoretical claim. Specifically, we show that Poisson SGD can learn parameters in a practical situation. Note that our aim is not to develop an effective method with high generalisation performance, but to develop a method that can evaluate the global convergence.

## 7.1 MNIST Dataset

We conducted experiments with the MNIST dataset (Deng, 2012). We consider a 4-layer fully connected neural network with 200 units of hidden layers are all 200 and the sigmoid activation function. We compare the performance of Poisson SGD with SGD, SGD with Momentum, SGLD. We set the batch size as 256, the learning rate of the SGD, the SGD with momentum and SGLD as 0.01, and the momentum coefficient as 0.9. We choose the hyper-parameter of Poisson SGD as $C_P = 100$, $C_\alpha = 100$, and $\beta = 10000$. We also use $\beta = 10000$ for SGLD. We use 60000 images for training and 10000 images for validation.

Figure 2 shows the result. The vertical line shows the misclassification ratio (%) and the horizontal line shows the number of epochs. We can see that Poisson SGD achieves sufficiently low errors, suggesting that it achieves a good minimum. In addition, Poisson SGD converges faster than other methods.

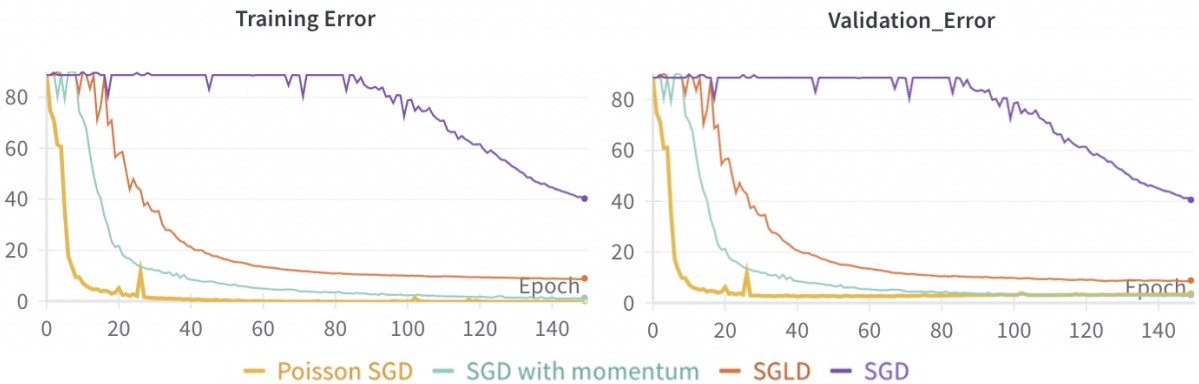

Figure 2: The comparison of the train error rate and valid error rate when optimized by Poisson SGD, SGD, SGD with momentum, and SGLD for 4-Layer DNN on MNIST.

## 7.2 CIFAR-10 Dataset

We conducted experiments with the CIFAR-10 dataset (Krizhevsky et al., 2009). We trained a convolutional neural network of 3 layers with the ReLU activation function, a $3 \times 3$ kernel, and the dropout rate 0.25. We compare the performance of Poisson SGD with SGD, SGD with Momentum, SGLD. We set the batch size as 256, the learning rate of SGD, SGD with momentum and SGLD as 0.01, and the momentum coefficient as 0.9. We choose the hyper-parameter of Poisson SGD as $C_P = 100$, $C_\alpha = 1$, and $\beta = 10000$. We also use $\beta = 10000$ for SGLD. We use 45000 data for training and 5000 data for validation.

Figure 3 shows the result. The vertical line shows the misclassification ratio (%) and the horizontal line shows the number of epochs. Although Poisson SGD is not the best method, it achieves sufficiently low errors, suggesting that it achieves a good minimum. In addition, it also achieves good accuracy comparable to that of SGD.

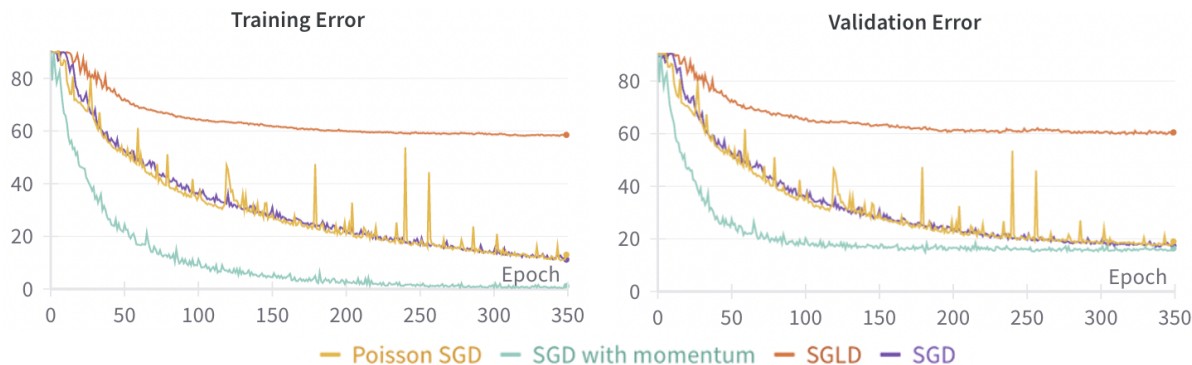

Figure 3: The comparison of the train error rate and valid error rate when optimized by Poisson SGD, SGD, SGD with momentum, and SGLD for CNN on Cifar-10.

## 8 Conclusion

We developed a new variant of SGD, Poisson SGD, whose search direction degenerates and derived its stationary distribution by incorporating a modification on the learning rate. The parameters trained by Poisson SGD are close enough to the global minima to take advantage of convergence to the stationary distribution. The generalization error is also evaluated. We believe that our work leads to the analysis of the actual SGD dynamics, not variants of it in the future.

**Broader Impact Statement**

The paper aims to provide a theoretical understanding of machine learning methods, which in itself has no direct negative impact on society.

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

## A  Supportive Information

We verify that the velocity vector is normalized by the choice of the momentum coefficient for Poisson SGD and BPS.

**Proposition 7.** *Consider the update* (4) *for $v_k$ with its momentum coefficient* (6). *Then, for $\forall k \in \{1, 2, ..., K\}$, we have $\|v_k\| = 1$. Further, for $\widehat{v}_k$ defined in Algorithm 2, we obtain $\|\widehat{v}_k\| = 1$ for every $k = 1, ..., K$.*

*Proof.* We first consider $v_k$ with the Poisson SGD case. Simply, we have

$$
\begin{aligned}
\|v_k\| &= \left\| v_{k-1} - 2\frac{\langle \nabla \widehat{L}_{\mathbf{z}}^{(k)}(\theta_k), v_{k-1} \rangle}{\|\nabla \widehat{L}_{\mathbf{z}}^{(k)}(\theta_k)\|^2} \nabla \widehat{L}_{\mathbf{z}}^{(k)}(\theta_k) \right\| \\
&= \left\| \left( I_d - 2\frac{\nabla \widehat{L}_{\mathbf{z}}^{(k)}(\theta_k) \nabla \widehat{L}_{\mathbf{z}}^{(k)}(\theta_k)^\top}{\|\nabla \widehat{L}_{\mathbf{z}}^{(k)}(\theta_k)\|^2} \right) v_{k-1} \right\| \\
&= \sqrt{v_{k-1}^\top \left( I_d - 2\frac{\nabla \widehat{L}_{\mathbf{z}}^{(k)}(\theta_k) \nabla \widehat{L}_{\mathbf{z}}^{(k)}(\theta_k)^\top}{\|\nabla \widehat{L}_{\mathbf{z}}^{(k)}(\theta_k)\|^2} \right)^2 v_{k-1}} \\
&= \|v_{k-1}\|.
\end{aligned}
$$

Since we set $\|v_0\| = 1$ for initialization, the statement holds.

For $\widehat{v}_k$ with the BPS case, the reflection does not change the norm of $\widehat{v}_k$ in the same way, and the refreshment also keeps $\|\widehat{v}_k\| = 1$, which completes the proof. □

## B  Proof of Theorem 1

*Proof.* By Theorem 3 and 4, we can bound the approximation error

$$
\mathcal{W}_1(\mu_{\mathbf{z},K}, \widehat{\mu}_{\mathbf{z},K}) \leq 4\sqrt{d}K\varepsilon,
$$

and the convergence error of BPS as

$$
\|\widehat{\mu}_{\mathbf{z},K} - \mu_{\mathbf{z}}^{(\beta,\varepsilon)}\|_{\mathrm{TV}} \leq \kappa(\beta, \varepsilon, d)^K.
$$

From Theorem 4 in Gibbs & Su (2002) (explicit form is Theorem 13 in Appendix H), we can bound the Wasserstein distance by the total variation, then obtain

$$
\mathcal{W}_1(\widehat{\mu}_{\mathbf{z},K}, \mu_{\mathbf{z}}^{(\beta,\varepsilon)}) \leq W\|\widehat{\mu}_{\mathbf{z},K} - \mu_{\mathbf{z}}^{(\beta,\varepsilon)}\|_{\mathrm{TV}} \leq W\kappa(\beta, \varepsilon, d)^K.
$$

The triangle inequality completes the proof. □

## C  Proof of Theorem 3

*Proof.* From the definition of Wasserstein distance,

$$\mathcal{W}_1(\mu_{\mathbf{z},k}, \widehat{\mu}_{\mathbf{z},k}) = \inf_{\pi \in \Pi(\mu_{\mathbf{z},k}, \widehat{\mu}_{\mathbf{z},k})} \mathbb{E}_\pi[\|\theta_k - \widehat{\theta}_k\|_1]$$

holds, so we study the distance between $\theta_k$ and $\widehat{\theta}_k$ in terms of the norm $\|\cdot\|_1$. Since $\|v_k\| = \|v_{k-1}\| = \|\widehat{v}_k\| = \|\widehat{v}_{k-1}\| = 1$ holds by Proposition 7, we have

$$
\begin{aligned}
\mathbb{E}_\pi[\|\theta_k - \widehat{\theta}_k\|_1] &= \mathbb{E}_\pi[\|\theta_{k-1} + \eta_k v_{k-1} - (\widehat{\theta}_{k-1} + \widehat{\eta}_k \widehat{v}_{k-1})\|_1] \\
&\leq \mathbb{E}_\pi[\|\theta_{k-1} - \widehat{\theta}_{k-1}\|_1] + \mathbb{E}_\pi[\|(\widehat{\eta}_k - \eta_k)\widehat{v}_{k-1} + \eta_k(\widehat{v}_{k-1} - v_{k-1})\|_1] \\
&\leq \mathbb{E}_\pi[\|\theta_{k-1} - \widehat{\theta}_{k-1}\|_1] + \mathbb{E}_\pi[\|(\widehat{\eta}_k - \eta_k)\widehat{v}_{k-1}\|_1] + \mathbb{E}_\pi[\|\eta_k(\widehat{v}_{k-1} - v_{k-1})\|_1] \\
&\leq \mathbb{E}_\pi[\|\theta_{k-1} - \widehat{\theta}_{k-1}\|_1] + \sqrt{d}\mathbb{E}_\pi[|\eta_k - \widehat{\eta}_k|] + 2\sqrt{d}\mathbb{E}_\pi[\eta_k],
\end{aligned}
\tag{11}
$$

where we use $\|\cdot\|_1 \leq \sqrt{d}\|\cdot\|$ in the last inequality.

We first evaluate the second term of (11). There exists a coupling $\pi$ such that

$$\mathbb{E}_\pi[|\eta_k - \widehat{\eta}_k|] = \mathcal{W}_1(P_{\eta_k}, P_{\widehat{\eta}_k})$$

holds, where $P_{\eta_k}$ and $P_{\widehat{\eta}_k}$ denote the distribution of $\eta_k$ and $\widehat{\eta}_k$ respectively. We use such a coupling as $\pi$. In evaluating $\mathcal{W}_1(P_{\eta_k}, P_{\widehat{\eta}_k})$, we consider the following analysis. $\eta_k$ and $\widehat{\eta}_k$ are 1-dimensional and their cumulative distribution function is written as

$$
F_1(t) = 1 - \exp\left(-\int_0^t (\beta\langle\nabla\widehat{L}_{\mathbf{z}}^{(k)}(\theta + rv), v\rangle_+ + C_P)dr\right),
$$
$$
F_2(t) = 1 - \exp\left(-\int_0^t (\beta\langle\nabla L_{\mathbf{z}}(\theta + rv), v\rangle_+ + C_B + \Lambda_{\text{ref}})dr\right),
$$

respectively, and we also have

$$
\begin{aligned}
&\beta\langle\nabla\widehat{L}_{\mathbf{z}}^{(k)}(\theta + rv), v\rangle_+ + C_P \geq C_P, \\
&\beta\langle\nabla L_{\mathbf{z}}(\theta + rv), v\rangle_+ + C_B + \Lambda_{\text{ref}} \geq C_B + \Lambda_{\text{ref}}, \text{ and} \\
&|(\beta\langle\nabla\widehat{L}_{\mathbf{z}}^{(k)}(\theta + rv), v\rangle_+ + C_P) - (\beta\langle\nabla L_{\mathbf{z}}(\theta + rv), v\rangle_+ + C_B + \Lambda_{\text{ref}})| \\
&\quad \leq \max\{|-\beta M_\ell + C_P - (C_B + \Lambda_{\text{ref}})|, |\beta M_\ell + C_P - (C_B + \Lambda_{\text{ref}})|\}.
\end{aligned}
$$

Hence, we can use Lemma 8 and obtain

$$\mathcal{W}_1(P_{\eta_k}, P_{\widehat{\eta}_k}) \leq \frac{\max\{|-\beta M_\ell + C_P - (C_B + \Lambda_{\text{ref}})|, |\beta M_\ell + C_P - (C_B + \Lambda_{\text{ref}})|\}}{C_P(C_B + \Lambda_{\text{ref}})}. \tag{12}$$

Next, we evaluate the third term of (11). We have

$$
\begin{aligned}
\mathbb{E}[\eta_k] &= \int_0^\infty P(\eta_k \geq t)dt \\
&= \int_0^\infty \exp\left(-\int_0^t \{\beta\langle\nabla\widehat{L}_{\mathbf{z}}^{(k)}(\theta_{k-1} + rv_{k-1}), v_{k-1}\rangle_+ + C_P\}dr\right)dt \\
&\leq \int_0^\infty \exp(-C_P t)\,dt \\
&= \frac{1}{C_P}.
\end{aligned}
\tag{13}
$$

Substituting (12) and (13) into (11), we have

$$\mathbb{E}_\pi[\|\theta_k - \widehat{\theta}_k\|_1] \leq \mathbb{E}_\pi[\|\theta_{k-1} - \widehat{\theta}_{k-1}\|_1] \tag{14}$$

$$+\frac{\sqrt{d}\max\{|-\beta M_\ell + C_P - (C_B + \Lambda_{\text{ref}})|, |\beta M_\ell + C_P - (C_B + \Lambda_{\text{ref}})|\}}{C_P(C_B + \Lambda_{\text{ref}})} + \frac{2\sqrt{d}}{C_P}.$$

Since we take $C_P$ in Poisson SGD as $C_P = 1/\varepsilon$ and $C_B$ and $\Lambda_{\text{ref}}$ in BPS as $C_B + \Lambda_{\text{ref}} = \beta M_\ell + 1/\varepsilon$, (14) can be written as

$$\mathbb{E}_\pi[\|\theta_k - \widehat{\theta}_k\|_1] \leq \mathbb{E}_\pi[\|\theta_{k-1} - \widehat{\theta}_{k-1}\|_1] + 4\sqrt{d}\varepsilon.$$

Hence, solving this recursive inequality with $\theta_0 = \widehat{\theta}_0$, we have

$$\mathbb{E}_\pi[\|\theta_K - \widehat{\theta}_K\|_1] \leq 4\sqrt{d}K\varepsilon,$$

which is the desired conclusion. $\qquad\square$

**Lemma 8.** *Let $a_1$ and $a_2$ be $\mathbb{R}$-valued random variables whose cumulative distribution functions are*

$$F_1(t) = 1 - \exp\left(-\int_0^t f_1(r)dr\right), \text{ and } F_2(t) = 1 - \exp\left(-\int_0^t f_2(r)dr\right),$$

*respectively, where $f_1, f_2 : \mathbb{R} \to \mathbb{R}$ are continuous functions. Let the distributions of $a_1$ and $a_2$ be $P_1$ and $P_2$ respectively. Suppose that there exists $M, m_1, m_2 > 0$ such that $|f_2(t) - f_1(t)| \leq M$, $m_1 \leq f_1(t)$, and $m_2 \leq f_2(t)$ hold for $\forall t \in \mathbb{R}$. Then, the Wasserstein distance between $P_1$ and $P_2$ satisfies*

$$\mathcal{W}_1(P_1, P_2) \leq \frac{M}{m_1 m_2}.$$

*Proof.* Since $a_1$ and $a_2$ are 1-dimensional, we have

$$\mathcal{W}_1(P_1, P_2) = \int_0^1 \left|F_1^{-1}(q) - F_2^{-1}(q)\right| dq.$$

We introduce several notation $\delta(r) = f_2(r) - f_1(r)$, $t = F_1^{-1}(q)$, and $t' = F_2^{-1}(q)$, then

$$\int_0^t f_1(r)dr = \log\frac{1}{1-q}$$

$$\int_0^{t'} (f_1(r) + \delta(r))dr = \log\frac{1}{1-q}$$

holds. So, we obtain

$$\int_{t'}^t f_1(r)dr = \int_0^{t'} \delta(r)dr.$$

Hence, we have

$$\left|\int_{t'}^t f_1(r)dr\right| = \int_{\min\{t,t'\}}^{\max\{t,t'\}} f_1(r)dr \leq Mt'.$$

In addition, $\int_{\min\{t,t'\}}^{\max\{t,t'\}} f_1(r)dr \geq m_1|t - t'|$ holds, so we have

$$|t - t'| \leq \frac{Mt'}{m_1}.$$

We have the upper bound of $t'$ as

$$\log\frac{1}{1-q} = \int_0^{t'} f_2(r)dr \geq m_2 t',$$

so we have

$$|t - t'| \le \frac{M}{m_1 m_2} \log \frac{1}{1 - q}.$$

Since $\int_0^1 |\log(1 - q)| dq = 1$ holds, we obtain

$$\int_0^1 \left| F_1^{-1}(q) - F_2^{-1}(q) \right| dq \le \frac{M}{m_1 m_2}.$$

□

## D  Proof of Theorem 4

We prove this theorem by two steps. First, we prove that BPS has $\mu_\mathbf{z}^{(\beta, \varepsilon)}$ as one of its stationary distributions in section D.1. At this stage, BPS may have other forms of stationary distribution or may not converge to its stationary distribution. Second, we prove that BPS has a unique stationary distribution and converges to its stationary distribution at exponential rate, in other words, it has the exponential ergodicity, in section D.2.

### D.1  The form of the stationary distribution

In this section, we check that BPS has $\mu_\mathbf{z}^{(\beta, \varepsilon)}$ as a stationary distribution. In the proof, we define $\lambda(\theta, v) :=$ $\beta \langle \nabla L_\mathbf{z}(\theta), v \rangle_+, \bar{\lambda}(\theta, v) := \lambda(\theta, v) + \Lambda_{\text{ref}}$, and $R_\mathbf{z}(\theta) := I_d - 2 \frac{\nabla L_\mathbf{z}(\theta) \nabla L_\mathbf{z}(\theta)^\top}{\|\nabla L_\mathbf{z}(\theta)\|^2}$. We remark that $R_\mathbf{z}$ is a symmetric matrix and satisfies $R_\mathbf{z}(\theta)^2 = I_d$, so it is also an orthogonal matrix.

From the proof of Lemma 1 in the supplementary material of Deligiannidis et al. (2019), we can write the transition probability $\widehat{Q}$ of BPS as following for arbitrary measurable sets $A \subset \Theta$ and $B \subset \mathbb{S}^{d-1}$:

$$\begin{aligned} \widehat{Q}((\theta, v), A \times B) = \int_0^\infty \exp\left\{ -\int_0^s \left( \bar{\lambda}(\theta + uv, v) + C_B \right) du \right\} \\ \times \left( \bar{\lambda}(\theta + uv, v) + C_B \right) K((\theta + sv, v), A \times B) ds, \end{aligned} \tag{15}$$

where a transition kernel $K$ is expressed as

$$\begin{aligned} K((\theta, v), A \times B) = &\frac{\lambda(\theta, v) + C_B}{\bar{\lambda}(\theta, v) + C_B} \mathbb{1}[\theta \in A] \mathbb{1}[R_\mathbf{z}(\theta) v \in B] \\ &+ \frac{\Lambda_{\text{ref}}}{\bar{\lambda}(\theta, v) + C_B} \mathbb{1}[\theta \in A] \mu_{\text{unif}}(B), \end{aligned} \tag{16}$$

where $\mu_{\text{unif}}$ is the uniform probability measure on $\mathbb{S}^{d-1}$.

**Lemma 9.** *Under Assumption 1, a probability measure on $\Theta \times \mathbb{S}^{d-1}$*

$$\widehat{\mu}_z(A \times B) \propto \int_{A \times B} \left( \bar{\lambda}(\theta, -v) + C_B \right) \exp(-\beta L_z(\theta)) d\theta \mu_{\text{unif}}(dv)$$

*is the stationary distribution induced from the transition probability $\widehat{Q}$ as (15).*

*Proof.* Our proof is almost the same as the proof of Lemma 1 in Deligiannidis et al. (2019). Let $\pi_\mathbf{z}(d\theta, dv) = \exp(-\beta L_\mathbf{z}(\theta)) d\theta \mu_{\text{unif}}(dv)$.

First, we prove

$$\int (\bar{\lambda}(\theta, v) + C_B) \pi_\mathbf{z}(d\theta, dv) K((\theta, v), A \times B) \propto \widehat{\mu}_\mathbf{z}(A \times B). \tag{17}$$

Substituting (16), the left side of (17) is rewritten as

$$\int \pi_\mathbf{z}(d\theta, dv)(\lambda(\theta, v) + C_B) \mathbb{1}[\theta \in A] \mathbb{1}[R_\mathbf{z}(\theta) v \in B] + \int \pi_\mathbf{z}(d\theta, dv) \Lambda_{\text{ref}} \mathbb{1}[\theta \in A] \mu_{\text{unif}}(B).$$

We consider changing the variable as $v' = R_{\mathbf{z}}(\theta)v$. Since $R_{\mathbf{z}}(\theta)^{-1} = R_{\mathbf{z}}(\theta)$ holds, we get $\lambda(\theta, R_{\mathbf{z}}(\theta)^{-1}v') = \lambda(\theta, -v')$. In addition, since $|\det(R_{\mathbf{z}}(\theta))| = 1$, and $\mu_{\text{unif}}(R_{\mathbf{z}}(\theta)^{-1}dv') = \mu_{\text{unif}}(dv')$ hold due to the rotational invariance of $\mu_{\text{unif}}$, we obtain

$$\int_{A \times B} \pi_{\mathbf{z}}(d\theta, dv')(\lambda(\theta, -v') + C_B) + \int_{A \times B} \pi_{\mathbf{z}}(d\theta, dv')\Lambda_{\text{ref}},$$

which is proportional to the right side of (17) from the definition of $\widehat{\mu}_{\mathbf{z}}$.

Second, we prove $\int \widehat{Q}((\theta, v), (dy, dw))\widehat{\mu}_{\mathbf{z}}(d\theta, dv) = \widehat{\mu}_{\mathbf{z}}(dy, dw)$. We have

$$\int \widehat{Q}((\theta, v), (dy, dw))\widehat{\mu}_{\mathbf{z}}(d\theta, dv)$$

$$\propto \int_0^\infty \exp\left(-\int_0^s \{\bar{\lambda}(\theta + uv, v) + C_B\}du\right)\{\bar{\lambda}(\theta + sv, v) + C_B\}$$

$$\times K((\theta + sv, v), (dy, dw))\{\bar{\lambda}(\theta, -v) + C_B\}\pi_{\mathbf{z}}(d\theta, dv)ds.$$

If we change $\theta$ as $t = \theta + sv$, then this integral becomes

$$\int_0^\infty \exp\left(-\int_0^s \{\bar{\lambda}(t + (u-s)v, v) + C_B\}du\right)\{\bar{\lambda}(t, v) + C_B\}$$

$$\times K((t, v), (dy, dw))\{\bar{\lambda}(t - sv, -v) + C_B\}\pi_{\mathbf{z}}(d\theta, dv)ds.$$

Since $L_{\mathbf{z}}(\theta)$ is absolutely continuous,

$$\exp(-\beta L_{\mathbf{z}}(t - sv)) = \exp\left(-\beta L_{\mathbf{z}}(t) - \int_0^s \lambda(t - wv, -v)dw + \int_0^s \lambda(t - wv, v)dw\right)$$

holds in the same way as Deligiannidis et al. (2019). Substituting it into $\pi_{\mathbf{z}}(dx, dv)$ and changing $u$ as $u - s = -w$,

$$\int_0^\infty \exp\left(-\int_0^s \{\bar{\lambda}(t - wv, -v) + C_B\}dw\right)\{\bar{\lambda}(t - sv, -v) + C_B\}ds$$

$$\times\{\bar{\lambda}(t, v) + C_B\}K((t, v), (dy, dw))\pi_{\mathbf{z}}(dt, dv)$$

holds. The first line can be calculated as $\left[-\exp\left(-\int_0^s \{\bar{\lambda}(t - wv, -v) + C_B\}dw\right)\right]_0^\infty = 1$, so it is equal to

$$\int \{\bar{\lambda}(t, v) + C_B\}K((t, v), (dy, dw))\pi_{\mathbf{z}}(dt, dv).$$

Using (17), it is proportional to $\widehat{\mu}_{\mathbf{z}}(dy, dw)$, which completes the proof. □

By the following proposition, we prove that $\mu_{\mathbf{z}}^{(\beta,\varepsilon)}$ is one of the stationary distributions of BPS. Recall that we defined $a_d := \Gamma(d/2)/(\sqrt{\pi}\Gamma(d/2 + 1/2))$.

**Proposition 10.** *The marginal distribution of the stationary distribution expressed in Lemma 9 is written as*

$$\widehat{\mu}_z(d\theta) \propto (\Lambda_{\text{ref}} + C_B + a_d\beta\|\nabla L_z(\theta)\|) \exp(-\beta L_z(\theta))d\theta.$$

*Hence, if we put $\Lambda_{\text{ref}}$ and $C_B$ as $\Lambda_{\text{ref}} + C_B = \beta M_\ell + 1/\varepsilon$, it corresponds to $\mu_z^{(\beta,\varepsilon)}$.*

*Proof.* We only need to integrate with $v$ the distribution $\widehat{\mu}_{\mathbf{z}}$ expressed in Lemma 9. We have

$$\widehat{\mu}_{\mathbf{z}}(d\theta) \propto \int_{v \in \mathbb{S}^{d-1}} (\Lambda_{\text{ref}} + C_B + \beta\langle\nabla L_{\mathbf{z}}(\theta), -v\rangle_+) \exp(-\beta L_{\mathbf{z}}(\theta))d\theta\mu_{\text{unif}}(dv)$$

$$=(\Lambda_{\text{ref}} + C_B) \exp(-\beta L_{\mathbf{z}}(\theta))d\theta + \exp(-\beta L_{\mathbf{z}}(\theta))d\theta\beta\mathbb{E}_{v\sim\mu_{\text{unif}}}[\langle\nabla L_{\mathbf{z}}(\theta), v\rangle_+].$$

We can calculate the expected value in the last term as

$$\mathbb{E}_{v\sim\mu_{\text{unif}}}[\langle\nabla L_{\mathbf{z}}(\theta), v\rangle_+] = \mathbb{E}_{v\sim\mu_{\text{unif}}}[\|\nabla L_{\mathbf{z}}(\theta)\|(\cos\phi)_+] = \|\nabla L_{\mathbf{z}}(\theta)\|\mathbb{E}_{v\sim\mu_{\text{unif}}}[(\cos\phi)_+],$$

where $\phi \in \mathbb{R}$ is a random variable dependent on $v$ which satisfies

$$\cos\phi = \left\langle \frac{\nabla L_{\mathbf{z}}(\theta)}{\|\nabla L_{\mathbf{z}}(\theta)\|}, v \right\rangle. \tag{18}$$

From the symmetry of the uniform distribution, we can calculate $\mathbb{E}_{v\sim\mu_{\text{unif}}}[(\cos\phi)_+]$ by replacing $\frac{\nabla L_{\mathbf{z}}(\theta)}{\|\nabla L_{\mathbf{z}}(\theta)\|}$ in (18) by $(1, 0, \cdots, 0)$. Hence,

$$\mathbb{E}_{v\sim\mu_{\text{unif}}}[(\cos\phi)_+] = \mathbb{E}_{v\sim\mu_{\text{unif}}}[(v_1)_+] = \mathbb{E}\left[\left(\frac{x_1}{\sqrt{x_1^2 + \cdots + x_d^2}}\right)_+\right]$$

holds, where $v_1$ is the first component of $v$ and $x_i(i = 1, ..., d)$ is *i.i.d.* standard Gaussian variables.

For $(x_1, \ldots, x_d) \sim \mathcal{N}(\mathbf{0}, I_d)$, we have

$$\mathbb{E}\left[\sqrt{\frac{x_1^2}{x_1^2 + \cdots + x_d^2}}\right] = \int_{\mathbb{R}^d} \sqrt{\frac{z_1^2}{z_1^2 + \cdots + z_d^2}} \frac{1}{(2\pi)^{d/2}} \exp\left(-\frac{z_1^2 + \cdots + z_d^2}{2}\right) dz_1\cdots dz_d$$

$$= \int_{[0,\infty)^2} \sqrt{\frac{r}{r+s}} \frac{r^{-1/2}\exp(-r/2)}{\sqrt{2\pi}} \frac{s^{(d-1)/2-1}\exp(-s/2)}{\Gamma((d-1)/2)2^{(d-1)/2}} dr ds$$

$$= \int_{[0,1]} t^{1/2} \frac{t^{1/2-1}(1-t)^{(d-1)/2-1}}{\mathrm{B}(1/2, (d-1)/2)} dt$$

$$= \frac{\mathrm{B}(1, (d-1)/2)}{\mathrm{B}(1/2, (d-1)/2)}$$

$$= \frac{\Gamma(1)\Gamma((d-1)/2)\Gamma(d/2)}{\Gamma(1/2)\Gamma((d-1)/2)\Gamma(d/2+1/2)}$$

$$= \frac{\Gamma(d/2)}{\sqrt{\pi}\Gamma(d/2+1/2)}.$$

Note that for all $d \geq 2$,

$$\frac{1}{\sqrt{d/2}} \leq \frac{\Gamma(d/2)}{\Gamma(d/2+1/2)} \leq \frac{1}{\sqrt{d/2-1/2}}$$

holds (e.g., see Qi & Luo (2013)). Therefore, for all $d \geq 2$, we have

$$\mathbb{E}\left[\left(\frac{x_1}{\sqrt{x_1^2 + \cdots + x_d^2}}\right)_+\right] = \frac{\Gamma(d/2)}{2\sqrt{\pi}\Gamma(d/2+1/2)} \in \left[\frac{1}{\sqrt{2\pi d}}, \frac{1}{\sqrt{2\pi(d-1)}}\right].$$

$$\square$$

## D.2 The exponential ergodicity of BPS

The next proposition is on the minorization condition of the 2-skeletons of BPS on the restricted domains. In short, minorization means that the stochastic process can go from any measurable set to any measurable set in the parameter space, which is a sufficient condition for the exponential ergodicity in the compact parameter space. 2-Skeleton means 2 step of the stochastic process. This proposition completes the proof of Theorem 4.

**Proposition 11.** *Under Assumption 1, the 2-skeletons of BPS satisfies the minorization condition; that is, for some $c > 0$, for all $(\theta, v) \in \Theta \times \mathbb{S}^{d-1}$ and all measurable $E \subset \Theta \times \mathbb{S}^{d-1}$, we have*

$$\widehat{Q}^2((\theta, v), E) \geq c \int_\Theta \int_{\mathbb{S}^{d-1}} \mathbb{1}[(\theta, v) \in E] d\theta \mu_{\text{unif}}(dv).$$

*Moreover, BPS is exponentially ergodic in total variation distance.*

*Proof.* We partially follow the proof of Lemma 4 in Deligiannidis et al. (2019).

Let $f : \Theta \times \mathbb{S}^{d-1} \to [0, \infty)$ be a non-negative and bounded function. We also use the notation $M' = \sup_{(\theta,v) \in \Theta \times \mathbb{S}^{d-1}} (\bar{\lambda}(\theta, v) + C_B) < \infty$. By considering the event where the first update of $v$ is *refreshment* from $\text{Unif}(\mathbb{S}^{d-1})$, we see that for any $(\theta_0, v_0) \in \Theta \times \mathbb{S}^{d-1}$,

$$\int_{\Theta \times \mathbb{S}^{d-1}} f(\theta, v) \widehat{Q}^2((\theta_0, v_0), (d\theta, dv))$$

$$= \int_{\Theta \times \mathbb{S}^{d-1}} \int_{\Theta \times \mathbb{S}^{d-1}} f(\theta, v) \widehat{Q}((\theta_1, v_1), (d\theta, dv)) \widehat{Q}((\theta_0, v_0), (d\theta_1, dv_1))$$

$$\geq \frac{\Lambda_{\text{ref}}}{M'} \inf_{\theta_1 \in \Theta} \int_{\Theta \times \mathbb{S}^{d-1}} f(\theta, v) \widehat{Q}((\theta_1, v_1), (d\theta, dv)) \mu_{\text{unif}}(dv_1)$$

holds. We also obtain that for $T \sim \text{Exp}(M')$, $V_1, V_2 \sim^{\text{i.i.d.}} \text{Unif}(\mathbb{S}^{d-1})$, we have

$$\inf_{\theta_1 \in \Theta} \int_{\Theta \times \mathbb{S}^{d-1}} f(\theta, v) \widehat{Q}((\theta_1, v_1), d\theta dv) \mu_{\text{unif}}(dv_1)$$

$$\geq \inf_{\theta_1 \in \Theta} \frac{\Lambda_{\text{ref}}}{M'} \mathbb{E} \left[ \mathbb{1}[\theta_1 + TV_1 \in \Theta] f(\theta_1 + TV_1, V_2) \right]$$

$$\geq \inf_{\theta_1 \in \Theta} \frac{\Lambda_{\text{ref}}^2}{M'} \int_{[0, \infty) \times \mathbb{S}^{d-1}} \mathbb{1}[\theta_1 + tv_1 \in \Theta] e^{-M't} f(\theta_1 + tv_1, v) dt \mu_{\text{unif}}(dv_1) \mu_{\text{unif}}(dv)$$

$$\geq \inf_{\theta_1 \in \Theta} \frac{\Lambda_{\text{ref}}^2 e^{-M' \text{diam}(\Theta)}}{M'} \int_{[0, \infty) \times \mathbb{S}^{d-1}} \mathbb{1}[\theta_1 + tv_1 \in \Theta] f(\theta_1 + tv_1, v) dt \mu_{\text{unif}}(dv_1) \mu_{\text{unif}}(dv)$$

$$= \inf_{\theta_1 \in \Theta} \frac{\Lambda_{\text{ref}}^2 e^{-M' \text{diam}(\Theta)}}{M'} \int_{\Theta \times \mathbb{S}^{d-1}} \mathbb{1}[\theta \in \Theta] f(\theta, v) \|\theta - \theta_1\|^{1-d} d\theta \mu_{\text{unif}}(dv)$$

$$\geq \frac{\Lambda_{\text{ref}}^2 e^{-M' \text{diam}(\Theta)}}{M' \text{diam}(\Theta)^{d-1}} \int_{\Theta \times \mathbb{S}^{d-1}} f(\theta, v) d\theta \mu_{\text{unif}}(dv),$$

where the second last equality uses a change of coordinates. Since $f$ is generic, the minorization condition holds. Harris's theorem thus gives the exponential ergodicity of BPS. □

# E   Proof of Theorem 5

*Proof.* We prove in the same way as the proof of Theorem 2.1 in Raginsky et al. (2017). Let $\theta_\mu$ be a random variable satisfying $\theta_\mu \sim \mu_{\mathbf{z}}^{(\beta, \varepsilon)}$, where $\mu_{\mathbf{z}}^{(\beta, \varepsilon)}$ is defined in (7). We denote $\theta_K \sim \mu_{\mathbf{z}, K}$ as the output of Poisson SGD (Algorithm 1). We have

$$\mathbb{E}_{\mathbf{z}}[\mathbb{E}_{\theta_K}[L(\theta_K)]] - \inf_{\theta \in \Theta} L(\theta)$$

$$= \mathbb{E}_{\mathbf{z}}[\mathbb{E}_{\theta_K}[L(\theta_K)] - \mathbb{E}_{\theta_\mu}[L(\theta_\mu)]] + \{\mathbb{E}_{\mathbf{z}}[\mathbb{E}_{\theta_\mu}[L(\theta_\mu)]] - \inf_{\theta \in \Theta} L(\theta)\},$$

and the second term of right-hand side is written as

$$\mathbb{E}_{\mathbf{z}}[\mathbb{E}_{\theta_\mu}[L(\theta_\mu)]] - \inf_{\theta \in \Theta} L(\theta)$$

$$= \mathbb{E}_{\mathbf{z}}[\mathbb{E}_{\theta_\mu}[L(\theta_\mu)]] - \mathbb{E}_{\mathbf{z}}[\mathbb{E}_{\theta_\mu}[L_{\mathbf{z}}(\theta_\mu)]] + \left( \mathbb{E}_{\mathbf{z}}[\mathbb{E}_{\theta_\mu}[L_{\mathbf{z}}(\theta_\mu)]] - \inf_{\theta \in \Theta} L(\theta) \right).$$

Letting $\theta^\circ = \text{argmin}_{\theta \in \Theta} L(\theta)$, the second part of the right-hand side in the equation above is

$$\mathbb{E}_{\mathbf{z}}[\mathbb{E}_{\theta_\mu}[L_{\mathbf{z}}(\theta_\mu)]] - \inf_{\theta \in \Theta} L(\theta) = \mathbb{E}_{\mathbf{z}}[\mathbb{E}_{\theta_\mu}[L_{\mathbf{z}}(\theta_\mu)] - \inf_{\theta \in \Theta} L_{\mathbf{z}}(\theta)] + \left( \mathbb{E}_{\mathbf{z}}\left[ \inf_{\theta \in \Theta} L_{\mathbf{z}}(\theta) - L_{\mathbf{z}}(\theta^\circ) \right] \right)$$

$$\leq \mathbb{E}_{\mathbf{z}}[\mathbb{E}_{\theta_\mu}[L_{\mathbf{z}}(\theta_\mu)] - \inf_{\theta \in \Theta} L_{\mathbf{z}}(\theta)].$$

As a result, we have

$$\mathbb{E}_{\mathbf{z}}[\mathbb{E}_{\theta_K}[L(\theta_K)]] - \inf_{\theta \in \Theta} L(\theta) \leq \mathbb{E}_{\mathbf{z}}[\mathbb{E}_{\theta_K}[L(\theta_K)] - \mathbb{E}_{\theta_\mu}[L(\theta_\mu)]] \tag{19}$$

$$+ \mathbb{E}_{\mathbf{z}}[\mathbb{E}_{\theta_\mu}[L(\theta_\mu)] - \mathbb{E}_{\theta_\mu}[L_{\mathbf{z}}(\theta_\mu)]] \tag{20}$$

$$+ \mathbb{E}_{\mathbf{z}}[\mathbb{E}_{\theta_\mu}[L_{\mathbf{z}}(\theta_\mu)] - \inf_{\theta \in \Theta} L_{\mathbf{z}}(\theta)]. \tag{21}$$

To evaluate the terms (19), (20), and (21), we prepare the following lemma to calculate the upper bound of the difference between two expected value by the Wasserstein distance.

**Lemma 12.** *Consider probability measures $\mu$ and $\nu$ on $\Theta$. Suppose that $\sup_{z \in \mathcal{Z}} |\ell(z; 0)| \leq A$ and $\sup_{z \in \mathcal{Z}} \|\nabla \ell(z; 0)\| \leq B$ hold. Then, we obtain*

$$\left| \mathbb{E}_{\theta_1 \sim \mu}[\ell(z; \theta_1)] - \mathbb{E}_{\theta_2 \sim \nu}[\ell(z; \theta_2)] \right| \leq (c_1 W + B) \sqrt{W \mathcal{W}_1(\mu, \nu)}, \text{ and} \tag{22}$$

$$\left| \mathbb{E}_{\theta_1 \sim \mu}[L(\theta_1)] - \mathbb{E}_{\theta_2 \sim \nu}[L(\theta_2)] \right| \leq (c_1 W + B) \sqrt{W \mathcal{W}_1(\mu, \nu)}. \tag{23}$$

*Proof.* Under the assumption, Lemma 3.1 in Raginsky et al. (2017) holds. Hence, we have

$$\|\nabla \ell(z; \theta)\| \leq c_1 \|\theta\| + B, \forall \theta \in \Theta, \forall z \in \mathcal{Z} \tag{24}$$

$$\ell(z; \theta) \leq \frac{c_1}{2} \|\theta\|^2 + B\|\theta\| + A, \forall \theta \in \Theta, \forall z \in \mathcal{Z}. \tag{25}$$

Moreover, from Lemma 3.5 in Raginsky et al. (2017), for arbitrary two probability measures $\mu$ and $\nu$, if we let

$$\sigma^2 = \max\{\mathbb{E}_{\theta_1 \sim \mu}[\|\theta_1\|^2], \mathbb{E}_{\theta_2 \sim \nu}[\|\theta_2\|^2]\},$$

then we have

$$\left| \mathbb{E}_{\theta_1 \sim \mu}[\ell(z; \theta_1)] - \mathbb{E}_{\theta_2 \sim \nu}[\ell(z; \theta_2)] \right| \leq (c_1 \sigma + B) \mathcal{W}_2(\mu, \nu).$$

Obviously, it also holds that

$$\left| \mathbb{E}_{\theta_1 \sim \mu}[L(\theta_1)] - \mathbb{E}_{\theta_2 \sim \nu}[L(\theta_2)] \right| \leq (c_1 \sigma + B) \mathcal{W}_2(\mu, \nu).$$

Since we have $\sigma \leq W$ and $\mathcal{W}_2(\mu, \nu) = \inf_{\pi \in \Pi(\mu, \nu)} (\int_\Theta \|z - z'\|^2 d\pi(z, z'))^{1/2} \leq \inf_{\pi \in \Pi(\mu, \nu)} (\int_\Theta W \|z - z'\|_1 d\pi(z, z'))^{1/2} = \sqrt{W \mathcal{W}_1(\mu, \nu)}$, we obtain the statement. $\square$

We start evaluating each of the terms (19), (20), and (21).

First, we study (19). From (23) in Lemma 12, we have

$$\mathbb{E}_{\theta_K}[L(\theta_K)] - \mathbb{E}_{\theta_\mu}[L(\theta_\mu)] \leq (c_1 W + B) \sqrt{W \mathcal{W}_1(\mu_{\mathbf{z}, K}, \mu_{\mathbf{z}}^{(\beta, \varepsilon)})}$$

$$\leq (c_1 W + B) \sqrt{W d_K(\beta, \varepsilon, d)}. \tag{26}$$

Second, we evaluate (20) using the same approach as Raginsky et al. (2017). Here, we need to evaluate

$$\mathbb{E}_{\theta_\mu}[\ell(z; \theta_\mu)] - \mathbb{E}_{\theta_{\mu'}}[\ell(z; \theta_{\mu'})],$$

where $z \in \mathcal{Z}$ is an arbitrary sampled data, $\theta_{\mu'} \sim \mu_{\mathbf{z}'}^{(\beta,\varepsilon)}$ and $\mu_{\mathbf{z}'}^{(\beta,\varepsilon)}$ is the stationary distribution of BPS when one of the data $z_i$ is changed to arbitrary $\bar{z}_i \in \mathcal{Z}$ and $\mathbf{z}'$ is a dataset with replacing $z_i$ to $\bar{z}_i$, and $L_{\mathbf{z}'}$ be its corresponding empirical risk. From (22) in Lemma 12, we have

$$\mathbb{E}_{\theta_\mu}[\ell(z;\theta_\mu)] - \mathbb{E}_{\theta_{\mu'}}[\ell(z;\theta_{\mu'})] \leq (c_1 W + B) \mathcal{W}_2(\mu_{\mathbf{z}}^{(\beta,\varepsilon)}, \mu_{\mathbf{z}'}^{(\beta,\varepsilon)})$$

$$\leq (c_1 W + B) C_{\mu'} \left[ \sqrt{D(\mu_{\mathbf{z}}^{(\beta,\varepsilon)}||\mu_{\mathbf{z}'}^{(\beta,\varepsilon)})} + \left( \frac{D(\mu_{\mathbf{z}}^{(\beta,\varepsilon)}||\mu_{\mathbf{z}'}^{(\beta,\varepsilon)})}{2} \right)^{\frac{1}{4}} \right],$$

where $D(\cdot||\cdot)$ is KL-divergence and

$$C_{\mu'} := 2 \inf_{\lambda > 0} \left( \frac{1}{\lambda} \left( \frac{3}{2} + \log \int_\Theta e^{\lambda \|\theta\|^2} \mu_{\mathbf{z}'}^{(\beta,\varepsilon)}(d\theta) \right) \right)^{\frac{1}{2}},$$

which is from Corollary 2.3 in Bolley & Villani (2005) (explicit form is Theorem 14 in Section H). Also, since we have $\|\theta\| \leq W$, $C_{\mu'} \leq 2W$ holds. We denote the density functions of $\mu_{\mathbf{z}}^{(\beta,\varepsilon)}, \mu_{\mathbf{z}'}^{(\beta,\varepsilon)}$ as $p_{\mathbf{z}}, p_{\mathbf{z}'}$, and the normalization constants as $\Lambda_{\mathbf{z}}, \Lambda_{\mathbf{z}'}$ respectively. Let us calculate $D(\mu_{\mathbf{z}}^{(\beta,\varepsilon)}||\mu_{\mathbf{z}'}^{(\beta,\varepsilon)})$. We have

$$\frac{p_{\mathbf{z}}(\theta)}{p_{\mathbf{z}'}(\theta)} = \frac{\Lambda_{\mathbf{z}'}}{\Lambda_{\mathbf{z}}} \cdot \frac{\beta M_\ell + 1/\varepsilon + a_d \beta \|\nabla L_{\mathbf{z}}(\theta)\|}{\beta M_\ell + 1/\varepsilon + a_d \beta \|\nabla L_{\mathbf{z}'}(\theta)\|} \exp\left(-\beta(L_{\mathbf{z}}(\theta) - L_{\mathbf{z}'}(\theta))\right), \tag{27}$$

so in order to obtain the upper bound of $D(\mu_{\mathbf{z}}^{(\beta,\varepsilon)}||\mu_{\mathbf{z}'}^{(\beta,\varepsilon)})$, we suppress each of the three terms of the right-hand side of (27). First, we suppress the second term.

$$\|\nabla L_{\mathbf{z}}(\theta)\| = \left\| \nabla L_{\mathbf{z}'}(\theta) + \frac{1}{n}(\nabla \ell(z_i;\theta) - \nabla \ell(\bar{z}_i;\theta)) \right\|$$

$$\leq \|\nabla L_{\mathbf{z}'}(\theta)\| + \frac{1}{n}\|\nabla \ell(z_i;\theta) - \nabla \ell(\bar{z}_i;\theta)\|$$

$$\leq \|\nabla L_{\mathbf{z}'}(\theta)\| + \frac{2}{n}(c_1\|\theta\| + B),$$

where the last inequality is from (24). Hence,

$$\frac{\beta M_\ell + 1/\varepsilon + a_d \beta \|\nabla L_{\mathbf{z}}(\theta)\|}{\beta M_\ell + 1/\varepsilon + a_d \beta \|\nabla L_{\mathbf{z}'}(\theta)\|} \leq \frac{\beta M_\ell + 1/\varepsilon + a_d \beta \left( \|\nabla L_{\mathbf{z}'}(\theta)\| + \frac{2}{n}(c_1\|\theta\| + B) \right)}{\beta M_\ell + 1/\varepsilon + a_d \beta \|\nabla L_{\mathbf{z}'}(\theta)\|}$$

$$\leq 1 + \frac{2a_d \beta(c_1 W + B)}{n(\beta M_\ell + 1/\varepsilon)}$$

$$\leq 1 + \frac{2a_d(c_1 W + B)}{n M_\ell} \tag{28}$$

holds. Second, we suppress the third term. We have

$$\exp\left(-\beta(L_{\mathbf{z}}(\theta) - L_{\mathbf{z}'}(\theta))\right) = \exp\left(-\beta\left(\frac{1}{n}(\ell(z_i;\theta) - \ell(\bar{z}_i;\theta))\right)\right)$$

$$\leq \exp\left(\frac{\beta}{n}\left(\frac{c_1\|\theta\|^2}{2} + B\|\theta\| + A\right)\right)$$

$$\leq \exp\left(\frac{\beta}{n}\left(\frac{c_1 W^2}{2} + BW + A\right)\right), \tag{29}$$

where we use (25). Finally, we suppress the first term. Using (28) and (29), we have

$$\frac{\Lambda_{\mathbf{z}'}}{\Lambda_{\mathbf{z}}} = \frac{\int_{\theta \in \Theta} (\beta M_\ell + 1/\varepsilon + a_d \beta \|\nabla L_{\mathbf{z}'}(\theta)\|) \exp\left(-\beta L_{\mathbf{z}'}(\theta)\right) d\theta}{\int_{\theta \in \Theta} (\beta M_\ell + 1/\varepsilon + a_d \beta \|\nabla L_z(\theta)\|) \exp\left(-\beta L_z(\theta)\right) d\theta}$$

$$\leq \left(1 + \frac{2a_d(c_1 W + B)}{n M_\ell}\right) \exp\left(\frac{\beta}{n}\left(\frac{c_1 W^2}{2} + BW + A\right)\right). \tag{30}$$

Combining (28), (29) and (30), we have

$$\log \frac{p_{\mathbf{z}}(\theta)}{p_{\mathbf{z}'}(\theta)} \leq 2\log\left(1 + \frac{2a_d(c_1 W + B)}{n M_\ell}\right) + \frac{2\beta}{n}\left(\frac{c_1 W^2}{2} + BW + A\right)$$

$$\leq \frac{1}{n}\left(\frac{4a_d(c_1 W + B)}{M_\ell} + \beta(c_1 W^2 + 2BW + 2A)\right),$$

so

$$D(\mu_{\mathbf{z}}^{(\beta,\varepsilon)} || \mu_{\mathbf{z}'}^{(\beta,\varepsilon)}) \leq \frac{1}{n}\left(\frac{4a_d(c_1 W + B)}{M_\ell} + \beta(c_1 W^2 + 2BW + 2A)\right)$$

holds. We set $C_d = 4a_d(c_1 W + B)/M_\ell$ and $C = c_1 W^2 + 2BW + 2A$, then we have

$$(20) \leq 2W(c_1 W + B)\left(\left(\frac{C_d + \beta C}{n}\right)^{\frac{1}{2}} + \left(\frac{C_d + \beta C}{n}\right)^{\frac{1}{4}}\right). \tag{31}$$

Finally, we evaluate (21). Let us denote

$$\Lambda_{\mathbf{z}}(\theta) = \frac{\Lambda}{\beta M_\ell + 1/\varepsilon + a_d\beta\|\nabla L_{\mathbf{z}}(\theta)\|}$$

$$\Lambda = \int_{\theta \in \Theta} (\beta M_\ell + 1/\varepsilon + a_d\beta\|\nabla L_{\mathbf{z}}(\theta)\|)e^{-\beta L_{\mathbf{z}}(\theta)}d\theta.$$

Since the distribution of $\theta_\mu$ is

$$\mu_{\mathbf{z}}^{(\beta,\varepsilon)}(d\theta) \propto \left(\beta M_\ell + \frac{1}{\varepsilon} + a_d\beta\|\nabla L_{\mathbf{z}}(\theta)\|\right)\exp(-\beta L_{\mathbf{z}}(\theta)),$$

we have

$$\mathbb{E}_{\theta_\mu}[L_{\mathbf{z}}(\theta_\mu)] = -\frac{1}{\beta}\left(\mathbb{E}_{\theta_\mu}\left[\log\frac{e^{-\beta L_{\mathbf{z}}(\theta_\mu)}}{\Lambda_{\mathbf{z}}(\theta_\mu)}\right] + \mathbb{E}_{\theta_\mu}[\log\Lambda_{\mathbf{z}}(\theta_\mu)]\right)$$

$$= \frac{1}{\beta}\left(\mathbb{E}_{\theta_\mu}[-\log p_{\mathbf{z}}(\theta_\mu)] - \mathbb{E}_{\theta_\mu}[\log\Lambda_{\mathbf{z}}(\theta_\mu)]\right).$$

Since we have $\mathbb{E}_{\theta_\mu}[\|\theta_\mu\|^2] \leq W^2$, we can calculate the upper bound of $\mathbb{E}_{\theta_\mu}[-\log p_{\mathbf{z}}(\theta_\mu)]$ by the differential entropy of Gaussian distribution in the same way as the discussion of Section 3.5 in Raginsky et al. (2017):

$$\mathbb{E}_{\theta_\mu}[-\log p_{\mathbf{z}}(\theta_\mu)] \leq \frac{d}{2}\log\left(\frac{2\pi e}{d}W^2\right).$$

Using (24), we have

$$\log\Lambda_{\mathbf{z}}(\theta) \geq \log\frac{\Lambda}{\beta M_\ell + 1/\varepsilon + a_d\beta(c_1 W + B))}.$$

In addition,

$$\log\Lambda = \log\int_{\theta \in \Theta} (\beta M_\ell + 1/\varepsilon + a_d\beta\|\nabla L_{\mathbf{z}}(\theta)\|)e^{-\beta L_{\mathbf{z}}(\theta)}d\theta$$

$$\geq \log\int_{\theta \in \Theta} (\beta M_\ell + 1/\varepsilon)e^{-\beta L_{\mathbf{z}}(\theta)}d\theta$$

$$= \log(\beta M_\ell + 1/\varepsilon) + \log\int_{\theta \in \Theta} e^{-\beta L_{\mathbf{z}}(\theta)}d\theta$$

$$\geq \log(\beta M_\ell + 1/\varepsilon) - \beta L_{\mathbf{z}}^* + \frac{d}{2} \log \frac{2\pi}{c_1 \beta}$$

holds, where the last inequality is from the equation (3.21) in Raginsky et al. (2017). Here, we denote $L_{\mathbf{z}}^* = \inf_{\theta \in \Theta} L_{\mathbf{z}}(\theta)$. Hence, we have

$$
\begin{aligned}
(21) &\leq \frac{1}{\beta} \left( \frac{d}{2} \log \left( \frac{2\pi e}{d} W^2 \right) + \log \frac{\beta M_\ell + 1/\varepsilon + a_d \beta (c_1 W + B)}{\beta M_\ell + 1/\varepsilon} + \beta L_{\mathbf{z}}^* - \frac{d}{2} \log \frac{2\pi}{c_1 \beta} \right) - L_{\mathbf{z}}^* \\
&\leq \frac{1}{\beta} \left( \frac{d}{2} \log \frac{e W^2 c_1 \beta}{d} + \log \left( 1 + \frac{a_d (c_1 W + B)}{M_\ell} \right) \right).
\end{aligned}
\tag{32}
$$

We combine the result (26), (31), and (32), then obtain the statement. □

## F  Proof of Proposition 6

*Proof.* Let $\theta_\mu$ and $\theta_\nu$ be the random variable which obey the distributions $\mu_{\mathbf{z}}^{(\beta, \varepsilon)}$ and $\nu_{\mathbf{z}}^{(\beta)}$ respectively.

In the same way as Theorem 5, we have

$$\mathbb{E}_z[\mathbb{E}_{\theta_K}[L(\theta_K)]] - \inf_{\theta \in \Theta} L(\theta) \leq \mathbb{E}_z[\mathbb{E}_{\theta_K}[L(\theta_K)] - \mathbb{E}_{\theta_\mu}[L(\theta_\mu)]] \tag{33}$$

$$+ \mathbb{E}_z[\mathbb{E}_{\theta_\mu}[L(\theta_\mu)] - \mathbb{E}_{\theta_\nu}[L(\theta_\nu)]] \tag{34}$$

$$+ \mathbb{E}_z[\mathbb{E}_{\theta_\nu}[L(\theta_\nu)] - \mathbb{E}_{\theta_\nu}[L_{\mathbf{z}}(\theta_\nu)]] \tag{35}$$

$$+ \mathbb{E}_z[\mathbb{E}_{\theta_\nu}[L_{\mathbf{z}}(\theta_\nu)] - \inf_{\theta \in \Theta} L_{\mathbf{z}}(\theta)]. \tag{36}$$

(33) can be evaluated in the same as Theorem 5.

First, we evaluate (34). We have

$$\mathbb{E}_{\theta_\mu}[L(\theta_\mu)] - \mathbb{E}_{\theta_\nu}[L(\theta_\nu)] \leq W \mathcal{W}_2(\mu_{\mathbf{z}}^{(\beta, \varepsilon)}, \nu_{\mathbf{z}}^{(\beta)})$$

from the same discussion in the proof of Theorem 5. Since both $\theta_\mu$ and $\theta_\nu$ satisfy the log-Sobolev inequality, we can use Otto-Villani theorem (Bakry et al., 2014) (explicit form is Theorem 15 in Section H), and

$$\mathcal{W}_2(\mu_{\mathbf{z}}^{(\beta, \varepsilon)}, \nu_{\mathbf{z}}^{(\beta)}) \leq \sqrt{2 c_{\mathrm{LS}}^{(\beta)} D(\mu_{\mathbf{z}}^{(\beta, \varepsilon)} || \nu_{\mathbf{z}}^{(\beta)})}$$

holds, where $D$ denotes the KL-divergence and $c_{\mathrm{LS}}^{(\beta)}$ is the log-Sobolev constant of $\nu_{\mathbf{z}}^{(\beta)}$. We have

$$
\begin{aligned}
D(\mu_{\mathbf{z}}^{(\beta, \varepsilon)} || \nu_{\mathbf{z}}^{(\beta)}) &= \mathbb{E}_{\theta \sim \mu} \left[ \log \frac{(\beta M_\ell + 1/\varepsilon + a_d \beta \|\nabla L_{\mathbf{z}}(\theta)\|) \exp\left(-\beta L_{\mathbf{z}}(\theta)\right) / \Lambda_\mu}{\exp\left(-\beta L_{\mathbf{z}}(\theta)\right) / \Lambda_\nu} \right] \\
&\leq \mathbb{E}_{\theta \sim \mu} \left[ \log(\beta M_\ell + 1/\varepsilon + a_d \beta M_\ell) \frac{\Lambda_\nu}{\Lambda_\mu} \right],
\end{aligned}
$$

where $\Lambda_\mu$ and $\Lambda_\nu$ are normalizing constants of the density functions of $\mu_{\mathbf{z}}^{(\beta, \varepsilon)}$ and $\nu_{\mathbf{z}}^{(\beta)}$ respectively. We have

$$\frac{\Lambda_\nu}{\Lambda_\mu} = \frac{\int_\Theta \exp\left(-\beta L_{\mathbf{z}}(\theta)\right) d\theta}{\int_\Theta (\beta M_\ell + 1/\varepsilon + a_d \beta \|\nabla L_{\mathbf{z}}(\theta)\|) \exp\left(-\beta L_{\mathbf{z}}(\theta)\right) d\theta} \leq \frac{1}{\beta M_\ell + 1/\varepsilon},$$

hence we have

$$D(\mu_{\mathbf{z}}^{(\beta, \varepsilon)} || \nu_{\mathbf{z}}^{(\beta)}) \leq \log\left(1 + a_d \beta \varepsilon M_\ell\right).$$

As a result, we obtain

$$\mathbb{E}_{\theta_\mu}[L(\theta_\mu)] - \mathbb{E}_{\theta_\nu}[L(\theta_\nu)] \leq W \sqrt{2 c_{\mathrm{LS}}^{(\beta)} \log\left(1 + a_d \beta \varepsilon M_\ell\right)}. \tag{37}$$

Second, we evaluate (35). Let $v_{\mathbf{z}'}^{(\beta)}$ be the Gibbs distribution when one of the data $z_i$ is replaced by $z_i'$. In the same way as Section 3.6 in Raginsky et al. (2017), we have

$$\mathcal{W}_2(v_{\mathbf{z}}^{(\beta)}, v_{\mathbf{z}'}^{(\beta)}) \leq \frac{2c_{\mathrm{LS}}^{(\beta)} \beta M_\ell}{n}.$$

Hence, we have

$$\mathbb{E}_{\theta_v}[L(\theta_v)] - \mathbb{E}_{\theta_v}[L_{\mathbf{z}}(\theta_v)] \leq (c_1 W + B) \frac{2c_{\mathrm{LS}}^{(\beta)} \beta M_\ell}{n}. \tag{38}$$

Finally, we evaluate (36). This term can be evaluated on the same way as Proposition 3.4 in Raginsky et al. (2017) and we have

$$\mathbb{E}_{\theta_v}[L_{\mathbf{z}}(\theta_v)] - \inf_{\theta \in \Theta} L_{\mathbf{z}}(\theta) \leq \frac{1}{\beta} \left( \frac{d}{2} \log \left( \frac{2\pi e W^2}{d} \right) - \frac{d}{2} \log \frac{2\pi}{c_1 \beta} \right)$$

$$= \frac{d}{2\beta} \log \left( \frac{e W^2 c_1 \beta}{d} \right). \tag{39}$$

We combine the result (37), (38), and (39), then obtain the statement. □

## G   Proof of Theorem 2

*Proof.* Let $\theta_K, \theta_\mu$ be the random variables whose distribution is $\mu_{\mathbf{z},K}^{(\beta,\varepsilon)}$ and $\mu_{\mathbf{z}}^{(\beta,\varepsilon)}$ respectively. Let $L_{\mathbf{z}}^* = \min_{\theta \in \Theta} L_{\mathbf{z}}(\theta)$. We have

$$\mathbb{E}_{\theta_K}[L_{\mathbf{z}}(\theta_K)] - L_{\mathbf{z}}^* = (\mathbb{E}_{\theta_K}[L_{\mathbf{z}}(\theta_K)] - \mathbb{E}_{\theta_\mu}[L_{\mathbf{z}}(\theta_\mu)]) + (\mathbb{E}_{\theta_\mu}[L_{\mathbf{z}}(\theta_\mu)] - L_{\mathbf{z}}^*).$$

As the first term of the right-hand side, we can use the Wasserstein distance in the same way as the proof of Theorem 5 as in (26). Hence, we have

$$\mathbb{E}_{\theta_K}[L_{\mathbf{z}}(\theta_K)] - \mathbb{E}_{\theta_\mu}[L_{\mathbf{z}}(\theta_\mu)] \leq (c_1 W + B) \sqrt{W d_K(\beta, \varepsilon, d)}.$$

Further, using (32) in the Proof of Theorem 5,

$$\mathbb{E}_{\theta_\mu}[L_{\mathbf{z}}(\theta_\mu)] \leq \frac{1}{\beta} \left( \frac{d}{2} \log \frac{e W^2 c_1 \beta}{d} + \log \left( 1 + \frac{a_d(c_1 W + B)}{M_\ell} \right) \right) + L_{\mathbf{z}}^*$$

holds, which completes the proof. □

## H   Explicit citation of the existing theorems

**Theorem 13** (Theorem 4, Gibbs & Su (2002))**.** *On the compact set $\Omega$, the Wasserstein metric $d_W$ and the total variation distance $d_{TV}$ satisfy the following relation:*

$$d_W \leq \mathrm{diam}(\Omega) \cdot d_{TV},$$

*where* $\mathrm{diam}(\Omega) = \sup\{d(x, y) | x, y \in \Omega\}$.

**Theorem 14** (Corollary 2.3, Bolley & Villani (2005))**.** *Let $X$ be a measurable space equipped with a measurable distance $d$, let $p \geq 1$ and let $v$ be a probability measure on $X$. Assume that there exist $x_0 \in X$ and $\alpha > 0$ such that $\int e^{\alpha d(x_0,x)^p} dv(x)$ is finite. Then, $\forall \mu \in P(X)$,*

$$W_p(\mu, v) \leq C \left[ H(\mu|v)^{\frac{1}{p}} + \left( \frac{H(\mu|v)}{2} \right)^{\frac{1}{2p}} \right],$$

*where*

$$C = 2 \inf_{x_0 \in X, \alpha > 0} \left( \frac{1}{\alpha} \left( \frac{3}{2} + \log \int e^{\alpha d(x_0,x)^p} dv(x) \right) \right)^{\frac{1}{p}} < \infty.$$

**Theorem 15** (Theorem 9.6.1, Bakry et al. (2014)). *Let $\mu$ be a probability measure on $M$. If $\mu$ satisfies a logarithmic Sobolev inequality $LS(C)$ for some constant $C > 0$, then it satisfies following for every probability measure $\nu$ on $M$:*

$$\mathcal{W}_2(\mu, \nu)^2 \leq 2C \cdot D(\nu || \mu),$$

*where $\mathcal{W}_2$ denotes the Wasserstein-2 distance and $D$ denotes the Kullback-Leibler divergence.*

