# OpenReview forum: "Effect of Random Learning Rate: Theoretical Analysis of SGD Dynamics in Non-Convex Optimization via Stationary Distribution"
_TMLR — Rejected by TMLR_

### Review · Reviewer_EbXG · 2024-07-30

**Summary Of Contributions:**

This paper analyzes the distribution convergence of a SGD variant called Poisson SGD. Traditional SGD based algorithms do not converge to a stationary distribution as there are degenerate direrctions, i.e, directions which are not being explored. The authors propose Poisson SGD which utilizes a random learning rate which is sampled from an exponential distribution parameterized by gradient of loss and momentum vector. Poisson SGD converges to a stationary distribution even though there are degenerate directions. The authors derive the stationary distribution of poisson SGD, derived global convergence on the empirical risk, connects it with BPS and provides a generalization error analysis of Poisson SGD.

**Audience:**

Yes

**Broader Impact Concerns:**

None.

**Claims And Evidence:**

Yes

**Requested Changes:**

I would like the authors to answer the weakness points and incorporate the changes to these weaknesses.

**Strengths And Weaknesses:**

**Strengths**
The authors rigrorously derive the form of stationary distributioin of Poisson SGD. The assumptions are not strong such as there is no assumption on the gradient noise like in SGLD's analysis for stationary distribution. And the form of the stationary distribution makes intuitive sense based on the global geometry landscape.

**Weakness**
1) I think the expression "Poisson SGD degenerates" makes very liitle sense to me. The large random learning rate allows it to explore directions that were not explorable for a constant learning rate SGD. So, I do not see why the authors claim the Poisson SGD degenerates. Infact in their experimental figure-1, due to the possion sgd lr, the iterates can trace different minimas, which makes it not degenerate.

2) I was expecting some discussions on stability of Poisson SGD and it's learning rate effects. Since this is sampling from an exponential distribution, there is a small chance to sample a large learning rate causing the loss to diverge. Do the authors notice such occurences. This brings to my third point:

3) I did not find any experimental results on Poisson SGD for deep neural networks with toy datasets like MNIST and CIFAR-10. The experiment in figure-1 does very little justice to understand the practical effect of large learning rate.

4) Can the author discuss the similarity or even benefit of poisson-sgd compared to adaptive optimizers like rmsprop, adam etc. Since, these also utilizes an adaptive step-size based on the current gradient slope, I find there is a lot of similarity between gradient updates of both the methods near a local minima.

**Questions**

5) From equation 7, consider two global minimas having different curvature at the minima. The distribution in equation 7 can't distinguish between these two global minimas. Although, it is expected that poisson sgd would escape from the sharper minima more easily.

---

> ### Author Response · Authors · 2024-08-22
> **Thank you for the comments.**
>
> Thank you very much for your thoughtful comments. We will reply to you individually here.
>
> **C1** . I think the expression "Poisson SGD degenerates" makes very liitle sense to me. The large random learning rate allows it to explore directions that were not explorable for a constant learning rate SGD. So, I do not see why the authors claim the Poisson SGD degenerates. In fact in their experimental figure-1, due to the possion sgd lr, the iterates can trace different minimas, which makes it not degenerate.
>
> **A1**: Let us clarify the definition of "degeneracy." In this context, degeneracy refers to the phenomenon where the covariance matrix of gradient noise resulting from stochastic sampling experiences a rank deficiency. In fact, the covariance matrix of SGD's gradient noise can be represented as $1/m * \sum_{i=1}^m (\nabla \ell(x_i;\theta)-\nabla L_Z(\theta))(\nabla \ell(x_i;\theta)-\nabla L_Z(\theta))^\top$, where each term in the sum is a rank-1 matrix, leading to a total covariance matrix with a rank no greater than $m$. Therefore, when considering over-parameterized models like DNNs, the matrix becomes rank-deficient. Similarly, the covariance matrix of gradient noise in Poisson SGD also exhibits rank deficiency, and under this interpretation of degeneracy, it can be considered degenerate as well.
>
> We added a discussion based on this under Remark 2 in Section 2.
>
> ----
>
> **C2**: I was expecting some discussions on stability of Poisson SGD and it's learning rate effects. Since this is sampling from an exponential distribution, there is a small chance to sample a large learning rate causing the loss to diverge. Do the authors notice such occurences. This brings to my third point:
>
> **A2**: As stated in the main theorem, by taking the parameter of the exponential distribution to be sufficiently large, the probability of having a large learning rate becomes exceedingly low, thereby preventing divergence. More mathematically, the expected value and variance of the random variable (learning rate) that follows this distribution are finite, that is, we have $P(\eta\ge t)\le exp(-C_P t)$, so $E[\eta]\le\int_{0}^{\infty} t\exp(-C_P t)dt=1/C_P$ and $E[\eta^2]\le\int_{0}^{\infty} t^2\exp(-C_P t)dt=1/C_P^2$ holds.
>
> We added a discussion based on it to Section 3.
>
> ----
>
> **C3**: I did not find any experimental results on Poisson SGD for deep neural networks with toy datasets like MNIST and CIFAR-10. The experiment in figure-1 does very little justice to understand the practical effect of large learning rate.
>
> **A3**: We added experiments of DNN on MNIST and CNN on Cifar-10 at Section 7. As we state in the reply A2, the effect of the large learning rate does not appear in our case.
>
> ----
>
> **C4**: Can the author discuss the similarity or even benefit of poisson-sgd compared to adaptive optimizers like rmsprop, adam etc. Since, these also utilizes an adaptive step-size based on the current gradient slope, I find there is a lot of similarity between gradient updates of both the methods near a local minima.
>
> **A4**: Indeed, there is a similarity in that both methods are optimization techniques that adapt learning rates through gradients. However, RMSProp and Adam do not incorporate randomness, which makes them susceptible to getting stuck in local minima. In contrast, Poisson SGD is superior in this aspect due to its inherent randomness, which allows for the occasional sampling of larger learning rates, enabling it to escape from local minima.
>
> We added a discussion based on this point at Introduction.
>
> ----
>
> **C5**: From equation 7, consider two global minimas having different curvature at the minima. The distribution in equation 7 can't distinguish between these two global minimas. Although, it is expected that poisson sgd would escape from the sharper minima more easily.
>
> **A5**: This point can be explained by considering the probability of approaching the global solution. We consider the probability of existence around a flat minimum $\theta_1 \in \Theta$ and a sharp minimum $\theta_2 \in \Theta$, when we find that, due to the shape of the distribution, a measure of an $\epsilon$-neighborhood of $\theta_1$ is greater than that within an $\epsilon$-neighborhood of $\theta_2$. Hence, we can claim that Poisson SGD also tends to favor flat minima.
>
> We added a discussion based on this point at the end of Section 4.1.2.

---

### Review · Reviewer_7mzm · 2024-08-04

**Summary Of Contributions:**

The authors propose a variant of stochastic gradient descent (SGD) that introduces a random parametric model into the learning rate values. The new variant, which they term Poisson SGD, has a learning rate that follows an exponential distribution. They demonstrate that the parameters updated by Poisson SGD converge to a stationary distribution. They further show that Poisson SGD finds global minima in non-convex problems.

**Audience:**

Yes

**Broader Impact Concerns:**

No specific concern or negative ethical implications.

**Claims And Evidence:**

Yes

**Requested Changes:**

In the introduction, can you add citations to the claim that SGD leads to lower generalization errors than GD?

On the second page, in first paragraph, you state that “investigate the optimization of complex loss functions such as deep neural networks”
DNNs are not loss functions, Do you mean such as those used for training DNNs?

In the discussion about the Gaussianity of the noise, it is worth while discussing recent papers that demonstrate that the noise of SGD is actually heavy tailed an not Gaussian.

[1] Simsekli, et al. "A tail-index analysis of stochastic gradient noise in deep neural networks." International Conference on Machine Learning. PMLR, 2019.

[2] Battash, et al.. "Revisiting the Noise Model of Stochastic Gradient Descent." International Conference on Artificial Intelligence and Statistics. PMLR, 2024.

In the related work section, what are the differences between the proposed SGD variant and existing schemes that rely on a random learning rate? The current explanation is very brief and not detailed enough.




On page 4, you claim that a specific distribution for the noise of SGD is not realistic. Why? This contradicts the results in [1] and [2].

Many details are missing from Figure 1, and the black trajectory looks like GD and not SGD. What is the batch size?

Some citations are repetitive, for example Zhanxing Zhu et al.

**Strengths And Weaknesses:**

**Strengths**

The authors study an important problem in machine learning. Namely, SGD and its variants are among the core algorithms in NN training.

The authors provide a theoretical analysis of several properties of the proposed Poisson SGD.

The authors demonstrate that their new variant leads to convergence of the parameters to a stationary distribution even without assuming Gaussianity of the gradient noise.

Overall the paper is easy to follow, and the English level is satisfactory.



**Weaknesses**

Since this is a new variant of SGD, I expect to see some empirical evaluation that demonstrates the effectiveness of the new approach.

Namely, I would expect to see results demonstrating how it performs in practice on real datasets and how long it takes to converge using this optimizer.

Additionally, the empirical role or $\beta$ should be evaluated to demonstrate the sensitivity of the variant to this parameter.

The assumption about the $lim_{K\rightarrow \infty} \quad \kappa()=0$ isn’t clarified in the paper and needs more details.

---

> ### Author Response · Authors · 2024-08-22
> **Thank you for the comments.**
>
> Thank you very much for your thoughtful comments. We will reply to you individually here.
>
> **C1**: In the introduction, can you add citations to the claim that SGD leads to lower generalization errors than GD?
>
> **A1**: For example, the following papers compare the generalization performance of SGD and GD. For example, Figure 1 in [1] or Figure 1 in [2] report that SGD has better generalization performance. We add to the introduction an explanation of the superiority of SGD due to this experimental result.
>
> [1] Wu et al. “On the Noisy Gradient Descent that Generalizes as SGD”, ICML 2020.
>
> [2] Zhu et al. “The Anisotropic Noise in Stochastic Gradient Descent: Its Behavior of Escaping from Sharp Minima and Regularization Effects”, ICML 2019.
>
> ----
>
> **C2**: On the second page, in first paragraph, you state that “investigate the optimization of complex loss functions such as deep neural networks” DNNs are not loss functions, Do you mean such as those used for training DNNs?
>
> **A2**: As you say, we intended the loss function used for training deep neural networks. We will correct the description here.
>
> ----
>
> **C3**: In the discussion about the Gaussianity of the noise, it is worth while discussing recent papers that demonstrate that the noise of SGD is actually heavy tailed an not Gaussian.
>
> **A3**: Thank you for raising an important point. To our understanding, in previous studies  including these papers, gradient noise can be close to Gaussian in some cases, but it can follow other distributions with heavier tail probability such as the power distribution. What we can see from these studies is that there is no clear agreement that gradient noise follows a particular distribution. Based on this background, we do not consider it appropriate to identify a specific distribution of gradient noise (e.g. Gaussian distribution).
> Based on these arguments, we update our introduction.
>
> ----
>
> **C4**: In the related work section, what are the differences between the proposed SGD variant and existing schemes that rely on a random learning rate? The current explanation is very brief and not detailed enough.
>
> **A4**: Musso (2020) proposed a learning rate that follows a uniform distribution with the aim of analyzing the scheduling of the learning rate, and also described the behavior of SGD in this case with equations. Biller et al. (2021) proposed a method where each of the sub-neural networks is updated according to a random learning rate, motivated e.g. by the lottery hypothesis of neural networks, and showed experimentally that this achieves good performance.
>
> These studies and ours have several major differences. The first difference is the way the learning rate is designed. Our method considers Poisson processes, whereas existing methods consider uniform distributions and heterogeneous learning rates for each subneural network. The second difference is the objective of the study. We aim to evaluate global convergence, whereas existing studies aim at interpretability, speed of convergence, etc., and have very different motivations.
>
> Thus, even though they are the same random learning rate studies, their development methods and research objectives are very different.
>
> We added a description based on this point in Section 1.2.
>
> ----
>
> **C5**: On page 4, you claim that a specific distribution for the noise of SGD is not realistic. Why? This contradicts the results in [1] and [2].
>
> **A5**: As we discussed in A3, in previous studies  including these papers, gradient noise can be close to Gaussian in some cases, but it can follow other distributions with heavier tail probability such as the power distribution. What we can see from these studies is that there is no clear agreement that gradient noise follows a particular distribution. Based on this background, we do not consider it appropriate to identify a specific distribution of gradient noise (e.g. Gaussian distribution).
>
> We put this detail in Remark 1.
>
> ----
>
> **C6**: Many details are missing from Figure 1, and the black trajectory looks like GD and not SGD. What is the batch size?
>
> **A6**: As you pointed out, the black trajectory represents GD, and the red trajectory represents GD with a learning rate modification applied through Poisson SGD. Since this is simply the optimization of a quartic function, there is no batch size involved; the updates are based solely on the gradients of the sampled points. To avoid confusion, we will change the notation from SGD to GD.
>
> We will put this detail at the caption of Figure 1.
>
> ----
>
> **C7**: Some citations are repetitive, for example Zhanxing Zhu et al.
>
> **A7**: We fixed the part accordingly.
>
> ----
>
> **C8**: Empirical evaluation is needed.
>
> **A8**: We added experiments of DNN on MNIST and CNN on Cifar-10 in Section 7.

---

> ### Author Response · Authors · 2024-08-22
>
> ----
>
> **C9**: The assumption $\kappa^K\rightarrow0$ isn’t clarified.
>
> **A9**: We discuss a form of $\kappa(\beta,\varepsilon,d)$ of other related algorithms, although we could not achieve the explicit form of $\kappa(\beta,\varepsilon,d)$ of Poisson SGD.
> In the case of Langevin dynamics with the setting of [3], $\kappa(\beta,\varepsilon,d)$ is $\Omega(c_{LS} k\eta / \beta(\beta+d))$, where $c_{LS}$ is the logarithmic Sobolev constant.
> On the other hand, explicitly deriving $\kappa(\beta,\varepsilon,d)$ for a class of PDMP is a challenging task as described in [4][5], as well as that of Poisson-SGD.
>
> We added the remark about it in section 4.1.2.
>
> - [3] Maxim Raginsky, Alexander Rakhlin, and Matus Telgarsky. Non-convex learning via stochastic gradient langevin dynamics: a nonasymptotic analysis. In Conference On Learning Theory, 2017.
> - [4] Alain Durmus, Arnaud Guillin, and Pierre Monmarché. Geometric ergodicity of the bouncy particle sampler. Annals of Applied Probability, 30:2069–2098, 2020.
> - [5] George Deligiannidis, Alexandre Bouchard-Côté, and Arnaud Doucet. Exponential ergodicity of the bouncy particle sampler. The Annals of Statistics, 47:1268–1287, 2019.

---

### Review · Reviewer_DLpU · 2024-08-12

**Summary Of Contributions:**

Authors propose to use random step size, which conforms to the Poisson distribution, in stochastic gradient descent algorithm, and claims convergence to stationary distribution in such a setting. In particular, authors claim that with proper choice of parameters, the algorithm can find global optimum for nonconvex objectives and generalization error can go to zero.

**Audience:**

Yes

**Claims And Evidence:**

No

**Requested Changes:**

(1) I really do not think it makes any sense to make the assumption that both the function values $l(z; \boldsymbol{\theta})$ and the norm of gradients $\|| \nabla_{\boldsymbol{\theta}} l \||$ are bounded. In other words, it is unconventional to assume the parameter space $\Theta$ is bounded.
- Considering that the probability of adopting a large step size based on Poisson distribution, authors must establish very explicit and soundly why such boundedness properties can hold.
- The entire convergence argument is established upon the boundedness assumption, as key bounds such as equations (29), (30) and all equations with quantities $M_l$ and $B$. The cited work [Bertazzi, Bierkens and Dobson, 2022] is not a proper example about first-order algorithm convergence, because its target is really about samplers and Markov process itself on a subset of a finite dimensional vector space. Authors can investigate further extant literatures in the SGD communities and explain why this assumption makes sense.


(2) Experimental validation of the proposed algorithm is essential and indispensable.

[Stochastic Gradient Langevin Dynamics with Variance Reduction, Z Huang, S Becker, 2021 International Joint Conference on Neural Networks (IJCNN)] might be an easier baseline to benchmark with than [User-friendly guarantees for the Langevin Monte Carlo with inaccurate gradient, Arnak S. Dalalyan and Avetik Karagulyane, Stochastic Processes and their Applications
Volume 129, Issue 12, December 2019, Pages 5278-5311], [Maxim Raginsky, COLT 2017] and [Faster Convergence of Stochastic Gradient Langevin Dynamics for Non-Log-Concave Sampling, Difan Zou, Pan Xu, Quanquan Gu Proceedings of the Thirty-Seventh Conference on Uncertainty in Artificial Intelligence, PMLR 161:1152-1162, 2021.]


(3) Comments about proof details:

- Page 18: Explain why $E_z E_{\theta_{\mu}} L_{z}(\theta_{\mu}) - E_{\theta_{\mu}}L_{z}(\theta_{\mu}) = 0 $.
- Important intermediate lemmas from external work should be cited explicitly: for example Corollary 2.3 in Bolley & Villani 2005 (page 20)

(4) To achieve the proclaimed benefits of global convergence and zero generalization gap, both of which are grand, authors should provide specific examples or give specific quantitative relationships about the hyperparameters $K, \delta, \beta, \varepsilon$.

**Strengths And Weaknesses:**

Weakness:

-- Main assumptions on boundedness of loss function values and the norm of its gradients are unconventional and unlikely to hold in realistic application scenarios.

-- Multiple central claims are vague, which I will detail in the Requested Changes section.

-- There is no experimental validation provided in the manuscript.

---

> ### Author Response · Authors · 2024-08-22
> **Thank you for the comments.**
>
> Thank you very much for your thoughtful comments. We will reply to you individually here.
>
> **C1**: I really do not think it makes any sense to make the assumption that both the function values  and the norm of gradients  are reasonable. In other words, it is unconventional to assume the parameter space  is bounded. Considering that the probability of adopting a large step size based on Poisson distribution, authors must establish very explicit and soundly why such boundedness properties can hold.
>
> **A1**: I apologize for the unclear presentation earlier. In this discussion, we consider the parameter space as a torus. This ensures that even with an unbounded learning rate, the parameters remain within the torus. While this setting may not be entirely realistic, it is widely used as a method to avoid the mathematically cumbersome procedures associated with bounded parameter spaces (such as projected gradient methods), as demonstrated in the following papers listed in A2. If one wishes to consider a bounded set in Euclidean space, the proposed algorithm can be adapted by adding a projection step. However, we believe that such procedures would complicate the analysis and obscure the main claims of the paper.
>
> Based on this comment, we put additional description at Section 2.1 to clearly present the setup with the torus.
>
> ----
>
> **C2**: The entire convergence argument is established upon the boundedness assumption. The cited work [Bertazzi, Bierkens and Dobson, 2022] is not a proper example about first-order algorithm convergence, because its target is really about samplers and Markov process itself on a subset of a finite dimensional vector space. Authors can investigate further extant literatures in the SGD communities and explain why this assumption makes sense.
>
> **A2**: We appreciate your comment. We believe that the boundedness of parameter spaces is an accepted setting in previous studies on SGD. For example, the following papers and books consider SGD or its variants with bounded parameter spaces (being compact convex sets or compact subsets of Riemannian manifolds).
>
> - Ljung, L. (1977). Analysis of recursive stochastic algorithms. IEEE transactions on automatic control, 22(4), 551—575.
> - Kushner, H., & Yin, G. (2003). Stochastic Approximation and Recursive Algorithms. Springer.
> - Bonnabel, S. (2013). Stochastic gradient descent on Riemannian manifolds. IEEE Transactions on Automatic Control, 58(9), 2217—2229.
> - Tripuraneni, N., Flammarion, N., Bach, F. & Jordan, M.I.. (2018). Averaging Stochastic Gradient Descent on Riemannian Manifolds. Proceedings of the 31st Conference On Learning Theory, in Proceedings of Machine Learning Research, 75:650—687.
> - Lan, G. (2020). First-order and Stochastic Optimization Methods for Machine Learning. Springer.
> - Boumal, N. (2023). An Introduction to Optimization on Smooth Manifolds. Cambridge University Press.
>
> Ljung (1977) explains that there is no unbounded implementation of SGD due to overflow, and Kushner and Yin (2003) argue that allowing unboundedness does not yield much comprehension of practical behaviors. These arguments motivate our settings.
>
> We added a description of this point in Section 2.1.
>
> ----
>
> **C3**: Experimental validation of the proposed algorithm is essential and indispensable.
>
> [Stochastic Gradient Langevin Dynamics with Variance Reduction, Z Huang, S Becker, 2021 International Joint Conference on Neural Networks (IJCNN)] might be an easier baseline to benchmark with than [User-friendly guarantees for the Langevin Monte Carlo with inaccurate gradient, Arnak S. Dalalyan and Avetik Karagulyane, Stochastic Processes and their Applications Volume 129, Issue 12, December 2019, Pages 5278-5311], [Maxim Raginsky, COLT 2017] and [Faster Convergence of Stochastic Gradient Langevin Dynamics for Non-Log-Concave Sampling, Difan Zou, Pan Xu, Quanquan Gu Proceedings of the Thirty-Seventh Conference on Uncertainty in Artificial Intelligence, PMLR 161:1152-1162, 2021.]
>
> **A3**: We added experiments of DNN on MNIST and CNN on Cifar-10 at Section 7. We tried SGLD with Variance Reduction as you mentioned, but our computing resource was not sufficient for calculating the full-batch gradient in our experiments on MNIST and Cifar-10, so we could not unfortunately.
>
> ----
>
> **C4**: Page 18: Explain why $E_z E_{\theta_\mu} L_z (\theta_\mu) - E_{\theta_\mu} L_z (\theta_\mu) = 0$.
>
> **A4**: We have not used this condition. Because our description was confusing, it may have misled readers into thinking that $E_zE_{θ_μ}[L(θ_μ)]-E_{θ_μ}[L_z(θ_μ)]$ is $E_z(E_{θ_μ}[L(θ_μ)]-E_{θ_μ}[L_z(θ_μ)])$. We have changed our description to prevent this confusion.
>
> ----
>
> **C5**: Important intermediate lemmas from external work should be cited explicitly: for example Corollary 2.3 in Bolley & Villani 2005 (page 20)
>
> **A5**: We added a description on several lemma invcluding the Corollary 2.3 in the additional section in Appendix, and cited them explicitly.

---

> ### Author Response · Authors · 2024-08-22
>
> **C6**: To achieve the proclaimed benefits of global convergence and zero generalization gap, both of which are grand, authors should provide specific examples or give specific quantitative relationships about the hyperparameters .
>
> **A6**: We cannot achieve the explicit form of $\kappa$, so the explicit condition in which $\kappa^K\rightarrow0$ cannot be attained. Although it is not directly related to the proposed method, in the case of Langevin dynamics, it has been determined that under the setting of [3], $\kappa$ is $\Omega(c_{LS} k\eta / \beta(\beta+d))$, where $c_{LS}$ is the logarithmic Sobolev constant. On the other hand, explicitly deriving $\kappa$ for Poisson-SGD is an extremely challenging task. For instance, even in [4][5], an explicit form of $\kappa$ is not provided. In the ideal situation for PDMP under limited circumstances, the explicit form of $\kappa$ has recently begun to be clarified ([6][7]). However, it remains highly challenging to provide an explicit form of $\kappa$ in complex settings similar to real-world SGD, like the ones we consider.
>
>
> We added a remark on this point after the statement.
>
> - [3] Maxim Raginsky, Alexander Rakhlin, and Matus Telgarsky. Non-convex learning via stochastic gradient langevin
> dynamics: a nonasymptotic analysis. In Conference On Learning Theory, 2017.
> - [4] Alain Durmus, Arnaud Guillin, and Pierre Monmarché. Geometric ergodicity of the bouncy particle sampler. Annals of Applied Probability, 30:2069–2098, 2020.
> - [5] George Deligiannidis, Alexandre Bouchard-Côté, and Arnaud Doucet. Exponential ergodicity of the bouncy particle sampler. The Annals of Statistics, 47:1268–1287, 2019.
> - [6] Christophe Andrieu, Alain Durmus, Nikolas Nüsken, and Julien Roussel. Hypocoercivity of piecewise deterministic Markov process-Monte Carlo. Annals of Applied Probability, 31(5): 2478-2517, 2021.
> - [7] Jianfeng Lu and Lihan Wang. On explicit L2-convergence rate estimate for piecewise deterministic Markov processes in MCMC algorithms. Annals of Applied Probability,  32(2): 1333-1361, 2022.

---

### Decision · Action_Editor_LUgM · 2024-10-03

**Recommendation:** Reject

**Comment:**

The reviewers found several issues with this submission and recommended rejecting it (2 out of 3). Specifically, they found that experimental results are not insufficiently explained and authors' rebuttal does not support their claim. The baselines against which authors benchmarked are simplified and experimental setting for provided baseline methods is unclearly stated. Reviewers also pointed several issues around assumptions, e.g., boundedness of the feasible and not involving a projection step. Reviewers found significant gap between theoretical bounds and experimental results.

**Audience:**

Yes, there is audience for this line of work.

**Claims And Evidence:**

The reviewers found several issues with this submission and recommended rejecting it. Specifically, they found that experimental results are not insufficiently explained and authors' rebuttal does not support their claim. The baselines against which authors benchmarked are simplified and experimental setting for provided baseline methods is unclearly stated. Reviewers also pointed several issues around assumptions, e.g., boundedness of the feasible and not involving a projection step. Reviewers found significant gap between theoretical bounds and experimental results.